# TRIM25 and ZAP target the Ebola virus ribonucleoprotein complex to mediate interferon-induced restriction

Rui Pedro Galão[1]*, Harry Wilson[1], Kristina L. Schierhorn[1], Franka Debeljak[1], Bianca S. Bodmer[2], Daniel Goldhill[3], Thomas Hoenen[2], Sam J. Wilson[4], Chad M. Swanson[1], Stuart J. D. Neil[1]*

1 Department of Infectious Diseases, School of Immunology and Microbial Sciences, King's College London, United Kingdom, 2 Institute for Molecular Virology and Cell Biology, Friedrich-Loeffler-Institut, Greifswald, Germany, 3 Section of Virology, Department of Medicine, Imperial College London, London, United Kingdom, 4 MRC Centre for Virus Research, University of Glasgow, United Kingdom

☯ These authors contributed equally to this work.
* rui_pedro.galao@kcl.ac.uk (RPG); stuart.neil@kcl.ac.uk (SJDN)

**Data Availability Statement:** All relevant data are within the manuscript and its Supporting Information files.

## Abstract

Ebola virus (EBOV) causes highly pathogenic disease in primates. Through screening a library of human interferon-stimulated genes (ISGs), we identified TRIM25 as a potent inhibitor of EBOV transcription-and-replication-competent virus-like particle (trVLP) propagation. TRIM25 overexpression inhibited the accumulation of viral genomic and messenger RNAs independently of the RNA sensor RIG-I or secondary proinflammatory gene expression. Deletion of TRIM25 strongly attenuated the sensitivity of trVLPs to inhibition by type-I interferon. The antiviral activity of TRIM25 required ZAP and the effect of type-I interferon was modulated by the CpG dinucleotide content of the viral genome. We find that TRIM25 interacts with the EBOV vRNP, resulting in its autoubiquitination and ubiquitination of the viral nucleoprotein (NP). TRIM25 is recruited to incoming vRNPs shortly after cell entry and leads to dissociation of NP from the vRNA. We propose that TRIM25 targets the EBOV vRNP, exposing CpG-rich viral RNA species to restriction by ZAP.

## Author summary

As part of the early host antiviral defence, RNA viruses such as Ebola Virus (EBOV) are sensed by pattern recognitions receptors (PRRs). PRR activation triggers signalling cascades that lead to the expression of type-I interferons (IFN), and the subsequent upregulation of hundreds of IFN-stimulated genes (ISGs), many of which have direct inhibitory activity on viral replication. Here we identified the E3 Ubiquitin-Ligase TRIM25 as an ISG potently antiviral against the replication of an EBOV transcription-and-replication-competent virus-like particle (trVLP) system. We demonstrated that TRIM25 interacts with viral proteins associated with incoming EBOV genome, promoting the ubiquitination of the viral nucleoprotein (NP), and its dissociation from the viral RNA. We showed that TRIM25 contributes for the IFN-mediated antiviral inhibition EBOV trVLP

**Funding:** These studies were funded by an MRC Discovery Award MC/PC/15068 and a Wellcome Trust Senior Research Fellowship (WT098049AIA) to SJDN, an MRC research grant MR/S000844/1 and a Guy's and St Thomas's Charity Challenge Fund grant to CMS and SJDN, and MRC research grant MR/M019756/1 to CMS. This project has received funding from the European Union's Horizon 2020 research and innovation programme under the Marie Skłodowska-Curie grant agreement No. 750621 (KLS). Additional funding was provided by the Friedrich-Loeffler-Institut as intramural funding (TH) and funding as part of the VISION consortium (BSB). The funders had no role in study design, data collection and analysis, decision to publish, or preparation of the manuscript.

**Competing interests:** The authors have declared no competing interests exist.

replication, and that this is dependent on another ISG protein, ZAP, which targets CpG dinucleotides in the viral genome. We suggest that TRIM25 targets the EBOV viral ribo-nucleoprotein in order to dissociate this structure and exposing the viral genome to the antiviral function of ZAP to limit viral replication.

## Introduction

Type I interferons (IFN-I) are released as part of the innate immune response to viruses and induce the expression of an array of genes, many of which have direct antiviral activity [1,2]. These proteins often target conserved structures or fundamental processes common to the replication of diverse viruses that cannot be avoided by simple point mutations in a given viral protein [3]. In turn, viruses encode mechanisms to inhibit either the detection of virally expressed pathogen associated microbial patterns (PAMPs) by pattern recognition receptors (PRR), or the signalling through the IFN-I receptor complex (IFNAR1/2). These strategies are essential for efficient *in vivo* viral propagation and pathogenesis [4,5]. However, viral evasion from innate immune responses is generally incomplete; acute viral infections are characterized by local and systemic IFN-I responses, and most viruses exhibit some degree of sensitivity to interferon treatment in their target cell types. Understanding the mechanisms by which IFN-I exerts its antiviral effects not only yields new understanding of how mammals have evolved to defend themselves against viral pathogens, but also reveals areas of vulnerability in virus replication that may be exploited therapeutically.

Ebola virus (EBOV, species *Zaire ebolavirus*) is the prototypic member of the ebolavirus genus of the *Filoviridae*, a family of filamentous enveloped non-segmented negative strand RNA viruses responsible for sporadic outbreaks in sub-Saharan Africa [6]. Ebola virus disease (EVD) is characterized by its high mortality rate (25%-90%), with symptoms including severe fever, vomiting, diarrhoea, that in some cases progress to haemorrhagic fever [7]. EBOV and other filoviruses are zoonotic infections in human, probably transmitted from bat species [8–10], and are spread from person to person by close contact with bodily fluids [11]. Individuals that recover from EVD can harbour persistent virus, and men can shed infectious EBOV in the semen for many months after viremia becomes undetectable [12–14].

EBOV replicates in various tissues but is thought to primarily infect cells of the myeloid lineage [15]. After uptake by macropinocytosis, the virion membrane fuses with the host endosome following the interaction between a proteolytically processed form of its glycoprotein (GP) and the cellular receptor, NPC1 [16–18]. Fusion releases the viral genome into the cytoplasm, allowing the initiation of the replication cycle [19]. The viral genomic RNA (vRNA) exists as a helical ribonucleoprotein complex (RNP) that contains the viral nucleoprotein (NP) and VP35, a co-factor of the viral RNA-dependent RNA polymerase L [20,21], and known chaperone of NP [22,23]. The RNP also contains at lower concentrations the polymerase L and the transcriptional regulator VP30 [24,25], while the RNP is further associated with VP24 [26], and surrounded by VP40. Finally, GP is embedded in the host cell-derived envelope. As with all negative-strand RNA viruses, the viral genome is first transcribed by L to generate mRNAs for viral gene products. L-mediated transcription initiates only at the 3' leader sequence of the vRNA. Viral genes are arrayed along the genome in an order correlating with their abundance in the infected cell; at the intergenic boundaries between each viral gene, L either dissociates from the genome, or reinitiates transcription, a process regulated by VP30 [19,27]. VP30 also acts as the molecular switch to promote the synthesis of a full length positive sense copy of the RNA genome (cRNA) [28,29]. The cRNA acts as template for the production of new copies of

vRNA, that are then assembled into new vRNPs within cytoplasmic inclusion bodies, which are themselves induced by vRNP components [30]. New vRNPs are then targeted to the plasma membrane (PM) where they interact with VP40, and bud from the surface as new virions [31,32].

EBOV encodes two major activities that counteract the innate immune responses. First, VP35 counteracts recognition of viral RNA species by the PRR retinoic acid induced gene I (RIG-I), as well as by playing an incompletely defined role in disabling proinflammatory signalling downstream of PRR activation [33–37]. Second, VP24 inhibits signalling through IFNAR1/2 by blocking nucleocytoplasmic transport of phosphorylated STAT1 [38,39]. Additionally, the virus may antagonize certain antiviral proteins directly; notably the activity of tetherin/BST2 is counteracted by EBOV GP in some experimental systems [40,41]. However, EBOV replication can still be inhibited by pre-treatment of target cells with IFN-I, and a number of ISGs have been proposed to have antiviral activities against EBOV [42–44]. Additionally, despite the known impairment of antiviral signalling by EBOV, upregulation of both type-I and type-II ISGs was observed in myeloid cells of EBOV-infected individuals [45], while elevated levels of IFN-beta [46] and significantly higher levels of IFN-alpha [47] were found in moderately ill and surviving patients suggesting that interferons are being stimulated in humans upon EBOV infection.

The requirement for high containment laboratories makes identifying host factors that facilitate or inhibit the replication of filoviruses challenging. To identify novel EBOV antiviral proteins, we used a well-established EBOV transcription- and replication-competent virus-like particle (trVLP) system to screen a human ISG library [48,49]. Amongst these, we identified tripartite-motif family member 25 (TRIM25) as a potent antiviral inhibitor of EBOV trVLP replication. TRIM25, an E3 ubiquitin ligase, until recently widely accepted as being a positive regulator of RIG-I-mediated pattern recognition of cytoplasmic viral RNA, has other emerging roles in innate immunity [50–52]. Of note, a recent study has implicated nuclear forms of TRIM25 as having direct antiviral activity against the influenza A virus vRNP [53]. Furthermore, TRIM25 is an important cofactor of the zinc-finger antiviral protein (ZAP, also known as ZC3HAV1 or PARP13), which broadly targets many RNA viruses [51,54]. Recent evidence has shown that ZAP binds to RNAs with a high content of CpG dinucleotides [55,56], targeting them for degradation. How TRIM25 regulates ZAP is unknown. Although TRIM25 interacts with and ubiquitinates ZAP, these events are not required for ZAP antiviral activity [51,54]. Here we show that TRIM25 interacts with the EBOV vRNP and induces the dissociation of NP from the genomic RNA, thereby promoting CpG-dependent restriction by ZAP.

## Results

### EBOV trVLP replication is sensitive to type I interferon

The EBOV trVLP system is a reverse genetics system that models the entire replication cycle of the virus without the need for high containment laboratory facilities (Fig 1A) [49]. This system involves a tetra-cistronic minigenome, cloned into a mammalian vector under a T7 polymerase promoter for initial transcription, which is flanked by the non-coding regions of EBOV, and encodes for a luciferase reporter, and the viral proteins VP40, GP, and VP24. trVLP stocks are produced in HEK293T cells (called passage 0 (p0) cells), by transfection of the tetra-cistronic minigenome, the auxiliary T7 polymerase and plasmids encoding the nucleocapsid components NP, VP35, VP30 and L. The produced trVLPs are then used to infect HEK293T or U87-MG target cells expressing the viral entry factor TIM1 (passage 1 (p1) cells). Propagation of trVLPs always requires the co-transfection of the vRNP components not expressed from the minigenome prior to infection. trVLP replication can be measured by Renilla

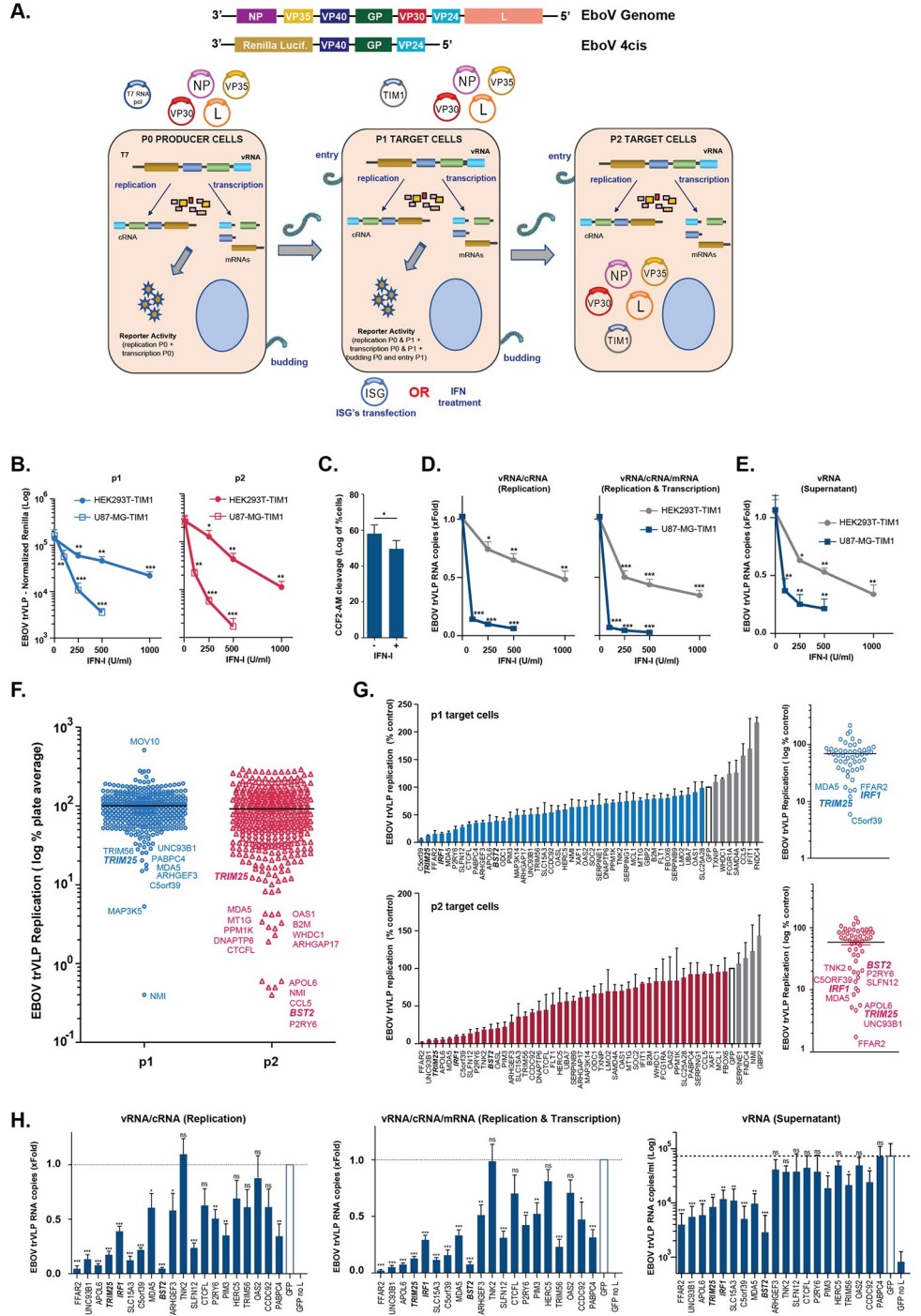

**Fig 1. Human ISG-expression screening identifies novel candidates with antiviral activity against EBOV trVLP.**
(**A**) Upper panels: Graphical representations of the Ebola virus genome, and of the tetra-cistronic minigenome (4cis) encoding Renilla luciferase (*Rluc*) used in the transcription- and replication-competent (trVLP) assay. Lower panel: Schematic representation of the EBOV trVLP assay. Plasmids expressing ISGs of interest were co-transfected with viral RNP proteins into p1 target cells. Alternatively, p1 target cells were pre-treated with IFN-I prior infection with EBOV trVLPs (detailed description in Material and Methods). (**B**) EBOV trVLP normalized reporter activity in HEK293T- and U87-MG-stably expressing TIM1 transfected with EBOV RNP proteins and pre-treated with increasing amounts of IFN-I 24 hours prior infection (p1 target cells, blue). Supernatants from p1 cells were harvested 24 hours post-infection, and used to infect HEK293T-TIM1 cells and reporter activities measured 24 hours later (p2 target cells, red).

**(C)** HEK293T-TIM1 cells were pre-treated with IFN-I (1000U/ml) prior to transduction with BlaVP40-EBOV-GP virus-like particles. 24 hours later, viral particle entry was measured as percentage of cells presenting cleavage of CCF2-AM dye by flow cytometry. **(D)** Relative quantification of intracellular viral RNA levels in HEK293T-TIM1 (grey) and U87-MG-TIM1 (blue) p1 target cells pre-treated with IFN-I, and infected with EBOV trVLPs as in (B). Random hexamer primers were used to generate cDNAs and RT-qPCR analysis was performed using qPCR primers/probe sets targeting the 5'-trailer region of the trVLP 4cis minigenome (vRNA and cRNA, left panel), VP40 RNA (vRNA, cRNA and mRNA, right panel) or *gapdh* as endogenous control. Data presented as fold change compared to control (no IFN) based on ΔΔCt values. **(E)** Relative quantification of viral transcripts present on supernatants from (D). cDNA synthesis performed as above and RT-qPCR performed with primers/probe set targeting EBOV VP40 RNA. Data shown as fold change compared to control (no IFN) based on absolute copy numbers. **(F)** Results of the arrayed human ISG screen. HEK293T-TIM1 target cells (p1) were pre-transfected with plasmids expressing individual ISGs together with EBOV RNP components and Firefly transfection control, and infected with EBOV trVLPs 24 hours later. Supernatants from p1 cells were harvested 24 hours post-infection and used to infect p2 target cells pre-transfected solely with vRNP components and pFluc. EBOV trVLP reporter activities for p1 (blue dots) and p2 target cells (red dots) were measured as in (B). Each dot represents one ISG. Infectivity measured for each ISG-expressing well was normalized to the activity of Fluc control within the well, and values are represented as percentage of the screen plate average, which is indicated as log of 100%. **(G)** Confirmatory assays for selected top candidate inhibitory ISGs. EBOV-trVLP infection was performed on HEK293T-TIM1 cells as in the primary screen. Normalized reporter activities for p1 (upper panels) and p2 target cells (lower panels) are represented as percentage of EBOV trVLP replication on cells transfected with GFP (white bars). **(H)** Relative quantification of intracellular (left and middle panels) and supernatant (right panel) viral RNA levels on HEK293T-TIM1 target cells (p1) transfected with EBOV RNP components and the top inhibitory ISG candidates and infected as before. cDNA synthesis and RT-qPCR analysis performed as in (D) and (E). All the represented EBOV trVLP Rluc reporter activities are normalized to control Fluc values obtained in the same lysates. Universal Type I IFN-α was used to pre-treat cells in panels (B-E). *p > 0.05, **p > 0.01 and ***p > 0.001 as determined by two-tailed paired t-test. All error bars represent ± SEM of at least three independent experiments.

luciferase (Rluc) activity in lysates of target cells p1, and by assessing infectivity of harvested p1 supernatants on new target TIM1 expressing HEK293T cells (passage 2 (p2) cells). p1 cells can also be used to assess the effects of exogenous treatments such as IFNs, or over-expression and knockdown/knockout of candidate cellular factors.

In order to identify candidate ISGs that inhibit EBOV replication, we first tested the sensitivity of trVLPs to IFN-I. HEK293T and U87-MG cells stably expressing TIM1 were pre-treated overnight with various doses of universal type-I IFN-α and then infected with equal doses of EBOV trVLPs. Rluc activity was measured in p1 cells 24h after infection, and trVLP yield was assessed in the supernatants by infecting HEK293T-TIM1 p2 cells. IFN-I treatment of both cell types gave a robust dose-dependent inhibition of Rluc activity in p1 and p2, although trVLP replication was more sensitive to IFN-I in U87-MG cells (Fig 1B). Given the distinct impact of IFN-α and IFN-ß on filovirus pathogenesis in animal models and humans [57], we tested their independent effect on EBOV trVLPs. As observed with the universal type-I IFN-α, both IFN-α2a and IFN-ß1a produced a dose-dependent inhibition of EBOV trVLPs on U87-MG cells, with IFN-ß1a being slightly more antiviral (S1A Fig), reproducing a previously reported observation [58]. Using an EBOV VP40-Blam VLP-based entry assay, IFN-I pre-treatment of target cells resulted in a minor reduction in entry of particles pseudotyped with EBOV-GP, indicating the majority of the inhibition of trVLP replication was taking place post viral entry (Fig 1C). In keeping with this, IFN-I treatment resulted in a dose-dependent reduction of vRNA and mRNA species in the infected cells (Fig 1D), as well as concomitant reduction of new viral minigenomes in the supernatant that mirrored the effects on Rluc activity (Fig 1E). Importantly, IFN-I treatment did not affect the levels of the vRNP proteins expressed *in trans* (S1B Fig). Thus, human cells exposed to type I IFN exhibit a robust inhibition of filoviral replication through the induction of one or more ISGs targeting trVLP gene expression and minigenome replication.

## A human arrayed ISG screen identifies TRIM25 as a potent inhibitor of EBOV trVLP replication

We then screened an expression library of 407 human ISGs individually cloned into mammalian expression vectors and measured the impact of each ISG on EBOV trVLP replication [48]. Briefly, individual expression vectors encoding a human ISG were transfected into HEK293T-TIM1 p1 cells alongside the vRNP components and a firefly luciferase (Fluc) transfection control. 24h later the cells were infected with a fixed dose of trVLPs equivalent to $10^6$ Rluc RLU/s on unmanipulated HEK293-TIM1 P1 cells. Rluc activity was measured at the harvest of supernatants 24h later. Supernatant infectivity was then assessed on p2 cells as before. Negative (GFP, empty vector) and positive (IRF1, BST2/tetherin) controls were included on each 96-well plate and Rluc activity in both p1 and p2 cells was normalized to the activity of control Fluc within individual wells and the effects of each individual gene was expressed as a percentage of the plate average (Fig 1F and S1 Table). Both p1 and p2 outputs of the screen were normally distributed (S1C Fig), however we reasoned that outlier ISGs that directly inhibited EBOV replication should always display a phenotype in p2 as well as potentially in p1, reflecting likely effects on early and/or late viral replication stages. We focussed on ISGs that gave at least a 5-fold reduction of trVLP infectivity in p2 and discarded any hit in p1 that did not carry over. The top 50 hits were validated in three independent experiments and ranked (Fig 1G). We then selected 20 of the top p2 hits from the validation for further characterization. First, we examined their effects on trVLP RNA using a primers-probe set in the *VP40* gene (that detects vRNA, cRNA and VP40 mRNA) and another set in the trVLP 5'-trailer region (that detects vRNA and cRNA) in p1 cell lysates, as well as measuring viral RNA levels in the supernatants as an indicator of released trVLP virions (Fig 1H). The majority of these ISGs, with the exception of TNK2, CTCFL, HERC5 and OAS2 affected the levels of cell-associated viral RNA. In addition to these, a few other ISGs had no significant impact on the viral RNA levels in the supernatants (ARHGEF3, SLFN12, P2RY6 and PABPC4). Several (FFAR2, UNC93B1, APOL6, SLC15A3, IFIH1 (MDA5), BST2, SLFN12 and PIM3) also reduced the expression of one or more of the viral RNP components expressed *in trans* (S1D Fig), potentially a confounding factor. Moreover, we noticed significant toxicity effects of FFAR2, C5ORF39, APOL6 and UNC93B1 consistent with previous reports of their proapoptotic activities (S1E Fig). Lastly, it is well known that many ISGs, and particularly pattern recognition receptors, can activate proinflammatory signalling pathways, indicating that antiviral activities measured could be mediated secondarily through upregulation of other ISGs or interferons. We therefore screened the selected 20 hits for their abilities to activate Fluc reporter genes driven by NFkB, IRF3 (ISG56/IFIT1) or IFN-induced ISGF3 (ISRE) dependent promoters (S1F Fig). As expected, expression of both IFIH1 (MDA5) and IRF1 activated ISG56 and ISRE promoters, indicating both an IRF3 and a secondary IFN-I-dependent response. IFIH1 (MDA5), FFAR2, P2YR6 and BST2 also triggered NFkB-dependent responses at both 24h and 48h. In contrast, TRIM25 only gave a detectable NFkB-dependent signal at 48h (i.e. 24h after p1 cells infection). Importantly, TRIM25 overexpression robustly inhibited the accumulation of trVLP RNA species in p1 cells without affecting expression of vRNP components or demonstrating overt toxicity (Figs 1H, S1D, and S1E). Moreover, recent evidence suggests that TRIM25 inhibits influenza virus independent of RIG-I [53]. Thus, TRIM25 was one of the candidate EBOV antiviral factors taken forward for further analysis.

## TRIM25 plays a major role in IFN-I-mediated inhibition of EBOV trVLP replication and reduces the abundance of viral RNA species

To further characterize the antiviral activity of TRIM25 we first showed that its ectopic expression in HEK293T-based p1 cells significantly blocked trVLP Rluc activity, both in these cells

and subsequent carry over to p2 at increasing trVLP inputs (Fig 2A), without affecting entry of EBOV-GP pseudotyped lentiviral vectors (Fig 2B). The block to trVLP replication correlated with reduced intracellular levels of viral mRNA, cRNA and vRNA, as well as vRNA levels in the supernatant (Fig 2C). We then used lentiviral-based CRISPR/Cas9 vectors to knockout TRIM25 in U87-MG and HEK293T cells (Fig 2D). In both cell types, TRIM25 is expressed at readily detectable levels and this expression is further augmented upon IFN-I treatment. Significantly, whereas IFN-I pre-treatment of control U87-MG cells carrying an irrelevant sgRNA targeting E.coli *lacZ* gene resulted in a 20–30 fold inhibition of trVLP infectivity in p1 target cells, knockout of TRIM25 significantly attenuated this block to trVLP production (Fig 2E), correlating with a rescue of viral RNA species in the corresponding cell lysates and supernatants (Fig 2F). Similarly, the antiviral effects of IFN-α2a or IFN-ß1a on EBOV trVLPs were also attenuated on U87-MG TRIM25 knockout cells (Fig 2G). Importantly, this effect of TRIM25 knockout was specific to the EBOV trVLPs, as IFN-I reduced the infectivity of VSV-G-pseudotyped HIV-1-based vectors equally in control sgRNA (*lacZ*) and knockout U87-MG cell lines (Fig 2H). Furthermore, in the absence of IFN treatment, TRIM25 knockout HEK293T cells gave significantly higher Rluc activity in p0 producer cells and yielded significant higher levels of virus (p1 infectivity) (Fig 2I). Thus, TRIM25 restricts trVLP production, and is an important component for the IFN-induced antiviral restriction of EBOV.

## TRIM25-mediated restriction of EBOV requires components of the cytoplasmic RNA pattern recognition machinery, but not proinflammatory signal transduction

The best characterized role of TRIM25 is as a cofactor of the cytoplasmic RNA sensor RIG-I [50]. TRIM25 and/or RIPLET mediate Lys63-linked polyubiquitination of RIG-I and promote the binding of this PRR to viral RNAs and its interaction with the full-length (FL) mitochondrial antiviral protein (MAVS) isoform [59,60]. This is essential for the propagation of proinflammatory signalling downstream of RIG-I and its related helicases [61]. We therefore used CRISPR/Cas9 to delete both RIG-I and MAVS in HEK293T and U87-MG cells (Fig 3A and 3B). Whilst RIG-I expression is only detectable after IFN-treatment of the control sgRNA cells (*lacZ*), FL-MAVS and a shorter isoform known as miniMAVS [62] are readily detected in both cell lines at steady state. Both isoforms are expressed as alternative translation products from the same mRNA. miniMAVS localizes and interacts with FL-MAVS, but having no CARD domains, cannot interact with RIG-like helicases and activate antiviral signal. Moreover, its interference with FL-MAVS activity has suggested it as a negative regulator of RNA sensing [62]. We designed sgRNA guides that would target exons upstream or downstream of the miniMAVS start codon and thus generated FL-MAVS knockout, or FL-MAVS and mini-MAVS double-knockout cells (DKO). We then examined the effects of these knockouts on the antiviral activity of ectopically expressed TRIM25. Whilst TRIM25 retained antiviral activity when overexpressed in RIG-I knockout cells it lost this activity in FL-MAVS/miniMAVS double-knockout cells (Fig 3C). Furthermore, there was no effect of RIG-I knock-out on the sensitivity of trVLPs to IFN-I in U87-MG cells (Fig 3D). By contrast, trVLP replication was significantly rescued from IFN-I antiviral effect in FL-MAVS/miniMAVS DKO cells, with FL-MAVS alone KO presenting only a partial rescue (Fig 3E). To clarify the potential requirement of each MAVS isoform for the antiviral activity of ectopically expressed TRIM25, we performed rescue experiments in which the expression of miniMAVS and/or FL-MAVS was stably reconstituted in HEK293T MAVS DKO cells (Fig 3F). The antiviral effect of TRIM25 against EBOV trVLP was rescued by the simultaneous reintroduction of CRISPR-resistant forms of miniMAVS and FL-MAVS (MAVS$^{CR}$), whilst still significantly attenuated when

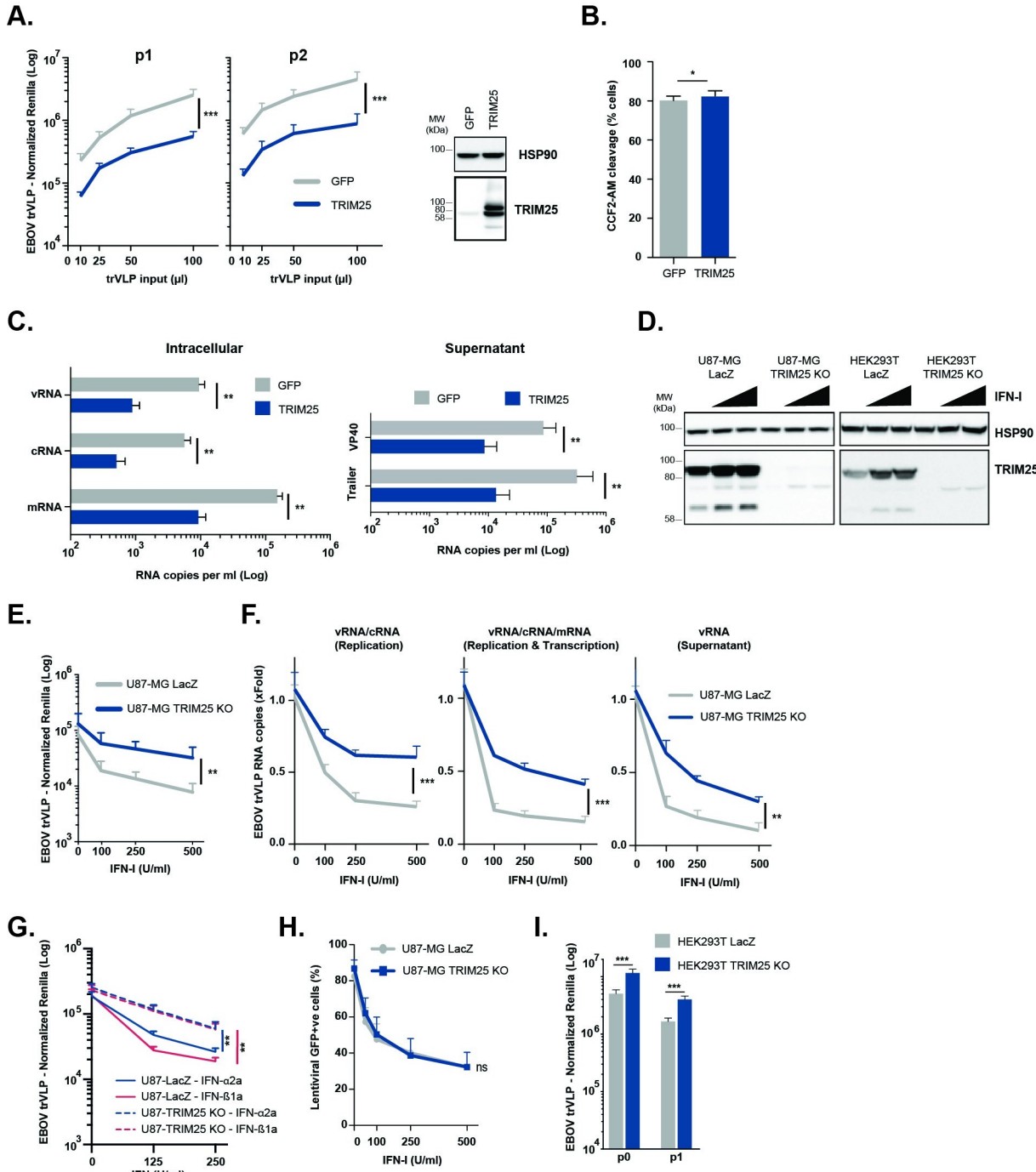

**Fig 2. TRIM25 is required for type-I IFN-mediated restriction of EBOV trVLP replication. (A)** EBOV trVLP normalized reporter activity on HEK293T-TIM1 cells transfected with EBOV RNP proteins and either GFP (grey) or TRIM25 (blue) prior to infection with increasing amounts of EBOV trVLPs (p1 target cells, left panel). Supernatants from p1 cells were harvested 24 hours post-infection and used to infect p2 target cells (right panel), and reporter activities measured 24 hours later. Protein levels of HSP90 and TRIM25 were determined by western blot on p1 cells at time of infection. **(B)** HEK293T-TIM1 cells were transfected with plasmids expressing either GFP (grey) or TRIM25 (blue), and transduced with BlaVP40-EBOV-GP virus like particles 24 hours later. Viral particle entry was determined 24 hours post-transduction by measuring the percentage of cells with cleaved CCF2-AM dye by flow cytometry. **(C)** Quantification of viral RNA transcripts present on cell lysates (left panel) and supernatants (right panel) of HEK293T-TIM1 cells transfected and infected as in (A). Strand-specific reverse transcription primers were used on total RNA extracted from cells to generate cDNAs for minigenomic RNA (vRNA), complementary RNA (cRNA), and mRNA, which were subsequently analysed by RT-qPCR. Random hexamer primers were used to generate cDNAs from total viral RNA extracted from supernatants, and qPCR analysis performed using primers/probe sets targeting 5' trailer region of the 4cis genome or

VP40 RNA. **(D)** U87-MG- and HEK293T-based CRISPR cells lines were treated with increasing amounts of IFN-I, and lysed 24 hours later for analysis. Protein levels of HSP90 and TRIM25 were determined by western blot on LacZ CRISPR control cells and corresponding TRIM25 CRISPR KO cell lines. **(E)** EBOV trVLP reporter activities on U87-MG LacZ CRISPR-TIM1 (grey) and U87-MG TRIM25 CRISPR KO-TIM1 (blue) target cells (p1) transfected with EBOV RNP proteins, and pre-treated with increasing amounts of IFN-I prior to infection. **(F)** Relative quantification of intracellular and supernatant trVLP RNA levels on U87-MG LacZ CRISPR-TIM1 (grey) and U87-MG TRIM25 CRISPR KO-TIM1 cells (blue) from (E). Random hexamer primers were used to generate cDNAs and RT-qPCR analysis was performed using primers/ probe sets targeting trVLP 4cis genome trailer region (vRNA and cRNA, left panel), or VP40 RNA (intracellular vRNA, cRNA and mRNA, middle panel; minigenomic RNA in the supernatant, right panel). Data are shown as fold change compared to control (no IFN) based on absolute copy numbers. **(G)** EBOV trVLP reporter activities on U87-MG LacZ CRISPR-TIM1 (solid lines) and U87-MG TRIM25 CRISPR KO-TIM1 (dashed lines) target cells transfected with EBOV RNP proteins and pre-treated with increasing amounts of IFN-α2a (blue) or IFN-ß1b (red) prior to infection. **(H)** U87-MG LacZ CRISPR (grey) and U87-MG TRIM25 CRISPR KO cells (blue) were treated with increasing amounts of IFN-I, and transduced the following day with a VSV-G pseudo-typed lentiviral vector expressing GFP (CSGW). The percentage of GFP-positive cells was determined 24 hours later by flow cytometry. **(I)** HEK293T LacZ CRISPR (grey) and HEK293T TRIM25 CRISPR KO (blue) were used as producer cells (p0) of EBOV trVLPs and reporter activities measured 48 hours post-transfection. Supernatants from p0 cells were used to infect HEK293T-TIM1 target cells (p1), and reporter activities determined 24 hours later. All the represented EBOV trVLP RLuc reporter activities are normalized to control FLuc values obtained in the same lysates. *$p > 0.05$, **$p > 0.01$ and ***$p > 0.001$ as determined by two-tailed paired t-test. All error bars represent ± SEM of at least three independent experiments.

FL-MAVS (M142A^CR) or miniMAVS (M1A^CR) were expressed separately (Fig 3G). Thus, the TRIM25 and IFN-I antiviral activities against EBOV trVLPs required both signalling active and inactive MAVS isoforms, but not RIG-I, suggesting a mechanism distinct from classical RIG-I-like receptors (RLR) signalling. To further rule out classical RNA sensing as the mechanism, we knocked out TBK1, which is the essential kinase downstream of MAVS for IFN and ISG induction (Fig 3H). As with RIG-I knockout, depletion of TBK1 did not affect TRIM25- or IFN-I-mediated inhibition of trVLP replication (Fig 3I and 3J). Moreover, TRIM25 overexpression in all knockout cell lines induced indistinguishable NFκB-dependent reporter activity at 48h irrespective of its antiviral effect, arguing against an indirect role of TRIM25-mediated signalling (Fig 3K). Taken together, these data indicate that the TRIM25/IFN-I-dependent antiviral restriction of EBOV trVLP replication requires both MAVS isoforms but is independent of the proinflammatory signal generated by classical cytoplasmic RLR sensing.

## Potent IFN-mediated inhibition of EBOV trVLP propagation requires ZAP and is modulated by CpG content of the viral genome

TRIM25 has been implicated in the regulation of ZAP, which has antiviral activity against numerous mammalian RNA viruses [51,54,63,64]. ZAP is expressed as long (L) and short (S) splice variants, the latter lacking the catalytically inactive PARP-like domain [65]. ZAP-L has more potent antiviral activities against some viruses and has a stronger interaction with TRIM25 than ZAP-S [51,65–67]. Whilst ZAP-S is the major IFN-regulated isoform [67,68] and was included in our ISG screen, it did not score as a hit. By contrast, however, overexpression of ZAP-L had a potent antiviral activity against EBOV trVLP replication in HEK293T-TIM1 p1 and p2 cells, which manifested as significant reductions in viral mRNA, vRNA and cRNA species (Figs 4A and S2A). We then generated HEK293T and U87-MG ZAP CRISPR knockout cells (Fig 4B), and found that like TRIM25 knockout cells, they exhibited a significantly reduced restriction of trVLP replication and viral RNA accumulation after pre-treatment with IFN-I (Fig 4C and 4D). We also observed an attenuated effect of IFN-α2a and IFN-ß1b on trVLP propagation (S2B Fig) and increased viral yields from p0 cells (S2C Fig). Furthermore, ZAP and TRIM25 interdependence was confirmed by showing that the antiviral effect conferred by their individual overexpression was abolished in cells lacking the other gene (Figs 4E and S2D). In contrast, while we could confirm the report [53] that overexpression of TRIM25 targets the influenza vRNP, the magnitude of this activity was unaffected by ZAP knockout (S2E Fig), suggesting mechanistic differences in EBOV trVLP restriction compared to previous observations made for Influenza.

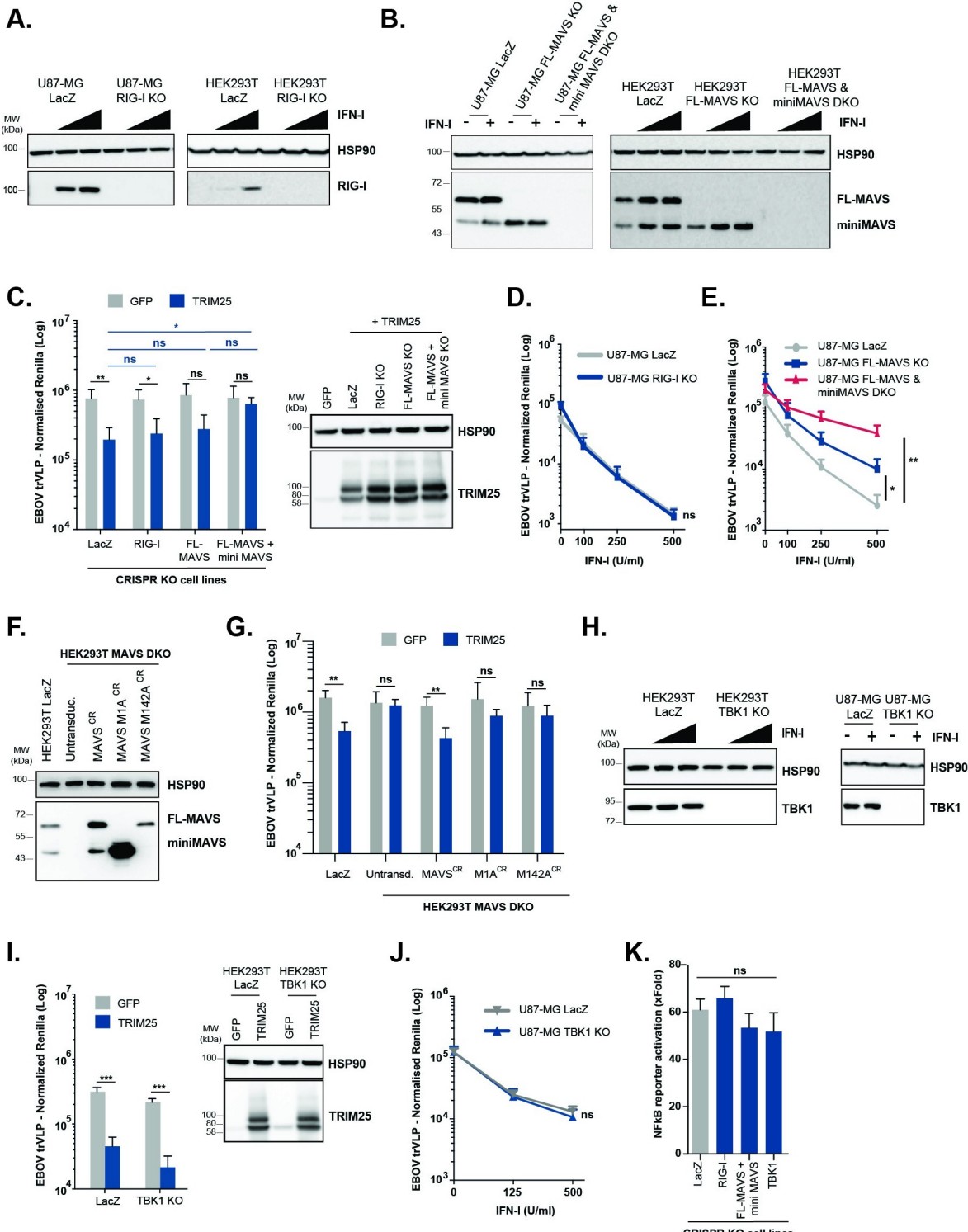

**Fig 3. TRIM25 antiviral effect against EBOV trVLPs is independent of RIG-I and downstream pro-inflammatory signal transduction. (A, B, H)** U87-MG- and/or HEK293T-based CRISPR cells lines were treated with increasing amounts of IFN-I, and lysed 24 hours later for protein analysis. Protein levels of HSP90 (A, B and H), RIG-I (A), FL-MAVS/mini-MAVS (B) and TBK1 (H) were determined by western blot on LacZ CRISPR control cells and corresponding CRISPR knock-out (KO) cells lines. **(C, I)** HEK293T-based CRISPR cell lines depicted in the figures were transfected with plasmids expressing EBOV RNP proteins and TIM1 together with either GFP (grey bars) or TRIM25 (blue bars), and later infected with a fixed amount of EBOV trVLPs. 24 hours post-infection cells were lysed

and trVLP reporter activities measured. Protein levels of HSP90 and TRIM25 were determined by western blot at time of infection. **(D)** EBOV trVLP reporter activities on U87-MG LacZ CRISPR (grey) and U87-MG RIG-I CRISPR KO (blue) target cells (p1), transfected with TIM1 and EBOV RNP proteins, and pre-treated overnight with increasing amounts of IFN-I prior to infection. Luciferase activities measured 24 hours post-infection. **(E)** EBOV trVLP reporter activities on U87-MG LacZ CRISPR (grey), U87-MG FL-MAVS KO (blue) and FL-MAVS/miniMAVS CRISPR DKO (red) target cells (p1), transfected with TIM1 and EBOV RNP proteins, and pre-treated overnight with increasing amounts of IFN-I prior to infection. Luciferase activities measured 24 hours post-infection. **(F)** Protein levels of HSP90 and MAVS determined by western blot lysates from HEK293T LacZ control cells, and HEK293T-MAVS DKO cell lines engineered to stably express CRISPR-resistant variants of both MAVS isoforms (MAVS$^{CR}$), miniMAVS (M1A$^{CR}$) or FL-MAVS (M142A$^{CR}$). **(G)** HEK293T LacZ CRISPR and engineered HEK293T-MAVS DKO cell lines from (F) were co-transfected with plasmids expressing EBOV RNP components and TIM1 together with either GFP (grey bars) or TRIM25 (blue bars), and later infected with EBOV trVLPs. Reporter activities were measured 24 hours later. **(J)** U87-MG LacZ CRISPR and U87-MG TBK1 CRISPR KO cells were transfected with RNP proteins and TIM1, followed by a IFN-I pre-treatment prior to infection with a fixed amount of EBOV trVLPs. EBOV trVLP reporter activities in p1 were measured 24 hours after infection. **(K)** Fold activation of a firefly luciferase NF-kB reporter in the depicted HEK293T-based CRISPR cells lines transiently transfected with TRIM25 compared to control GFP vector. Cells were harvested 48 hours post-transfection and FLuc reporter values normalised to control Renilla luciferase activity in the same lysates. All the represented EBOV trVLP Renilla reporter activities are normalized to control Firefly luciferase values obtained in the same lysates. $^{*}$p > 0.05, $^{**}$p > 0.01 and $^{***}$p > 0.001 as determined by two-tailed paired t-test. All error bars represent ± SEM of at least three independent experiments.

ZAP has previously been shown to inhibit the expression of EBOV proteins [69], particularly the RdRp L which we supply here *in trans*. However, whilst we could demonstrate significant reductions in L mRNA and protein levels in p1 cells transfected with ZAP-L (Figs 4F and S2F), the same reductions were observed in TRIM25 KO cells where ZAP overexpression had limited impact in reducing the levels of EBOV genomic RNA or antiviral activity (Figs 4E and S2G). Moreover, our earlier observations that IFN-I treatment did not affect the expression levels of vRNP components (S1D Fig) further argued against the reduction in L expression as the major contributor to the TRIM25/ZAP-mediated restriction of EBOV trVLPs under these experimental conditions.

Most mammalian viral RNA genomes exhibit marked suppression of CpG dinucleotides [56,70]. In the case of HIV-1, artificial increase in CpG concentration in the viral genome impairs replication through ZAP-mediated inhibition, with evidence that ZAP binds directly to the CpGs themselves and induces RNA degradation or translational repression [56]. The EBOV genome contains just over half of the number of CpGs expected by chance for its nucleotide composition (S2H Fig). These CpGs are widely distributed across the genome, both in the protein coding genes and intergenic regions with the notable exception of the intergenic region between GP and VP30 (S2I Fig). The trVLP CpG content is moderately higher, with 61 CpGs in the *Rluc* reporter gene that replaces NP as the first ORF. Since the *RLuc* coding sequence does not contain any necessary viral *cis*-acting regulatory elements, this gave us the opportunity to test whether CpG content could impact upon IFN-sensitivity of the virus. We generated a trVLP in genome in which all the CpGs in *Rluc* were silently mutated, reducing the observed vs expected CpG content of the trVLP to 0.45 (CpG low, S2H and S2I Fig). This modification had no detectable effect on trVLP yield or replication (Fig 4G). However, the CpG-low containing trVLP exhibited partial rescue from ZAP-L restriction (Fig 4G), which correlated with increased intracellular levels of viral mRNA, cRNA and vRNA in these cells, as well as vRNA levels in the supernatant (Fig 4H). Furthermore, the CpG-low containing trVLP was also less sensitive than the parental trVLP to IFN-I (Fig 4I). Since Renilla expression is not required for trVLP replication and both of these phenotypes were measured in p2 cells, these data imply that a major target of ZAP-mediated restriction is the CpG dinucleotides in the trVLP. These data further suggest that CpG content of the genomes of RNA virus can sensitize them to type I IFNs in a ZAP-dependent manner. Finally, ZAP-mediated targeting of CpG-containing retroviral RNA for degradation was shown to depend not only on TRIM25, but also on the putative endoribonuclease KHNYN, which predictably confers the nuclease activity that is lacking in both ZAP isoforms [71]. However, in contrast to ZAP-mediated restriction of

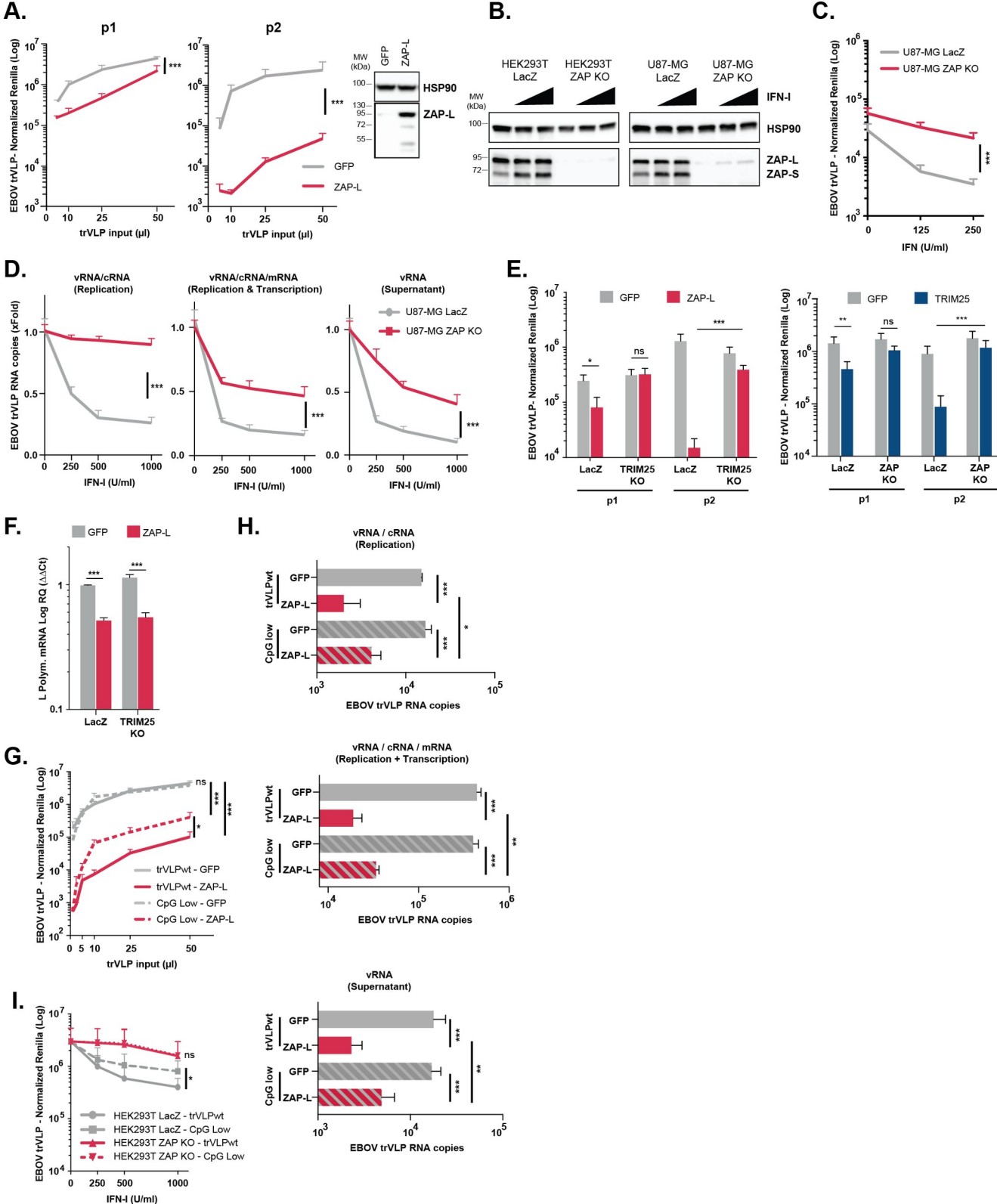

**Fig 4. TRIM25 and ZAP are inter-dependent for their antiviral activity against EBOV trVLP. (A)** EBOV trVLP normalized reporter activity on HEK293T-TIM1 cells transfected with EBOV RNP proteins and either GFP (grey) or ZAP-L (red) prior to infection with increasing amounts of EBOV trVLPs (p1 target cells, left panel). Supernatants from p1 cells were harvested and used to infect p2 target cells (middle panel), and reporter activities

measured 24 hours later. Protein levels of HSP90 and ZAP were determined by western blot at time of infection. **(B)** HEK293T- and U87-MG-based CRISPR cells lines were treated with increasing amounts of IFN-I, and lysed 24 hours later for analysis. Protein levels of HSP90 and ZAP were determined by western blot on LacZ CRISPR control cells and corresponding ZAP CRISPR KO cell lines. **(C)** EBOV trVLP reporter activities on U87-MG LacZ CRISPR-TIM1 (grey) and U87-MG ZAP CRISPR KO-TIM1 (red) target cells (p1), transfect with RNP proteins and pre-treated with increasing amounts of IFN-I prior to infection. Reporter activities measured 24 hours after infection. **(D)** Quantification of intracellular and supernatant viral RNA levels on U87-MG LacZ CRISPR-TIM1 (grey) and U87-MG ZAP CRISPR KO-TIM1 target cells (red) that were transfected with EBOV RNP proteins and pre-treated with IFN-I, prior to infection with EBOV trVLPs as in (C). Random hexamer primers were used to generate cDNAs and qPCR analysis was performed using primers/probe sets targeting either the trailer region of the viral genome (vRNA and cRNA, left panel), or VP40 RNA (intracellular vRNA, cRNA and mRNA, middle panel; minigenomic RNA in the supernatant, right panel). Data presented as fold change compared to control (no IFN) based on absolute copy numbers. **(E)** EBOV trVLP normalized reporter activities on HEK293T LacZ CRISPR, HEK293T TRIM25 CRISPR KO and HEK293T ZAP CRISPR KO cells stably expressing TIM1, that were transfected with EBOV RNP plasmids together with GFP (grey), ZAP L (red) or TRIM25 (blue), prior to infection with a fixed amount of EBOV trVLPs (p1 target cells). Supernatants from p1 were then harvested and used to infect HEK293T-TIM1 cells (p2 target cells). Protein levels of HSP90, TRIM25 and ZAP at time of infection shown in S4D Fig. **(F)** Relative quantification of EBOV L-Polymerase RNA transcripts on cell lysates of HEK293T LacZ CRISPR-TIM1 and HEK293T TRIM25 CRISPR KO-TIM1 cells transfected EBOV RNP plasmids in combination with either with GFP (grey) or ZAP-L (red), and infected with a fixed amount of EBOV trVLPs. Random hexamer primers were used to generate cDNAs from total RNA, and RT-qPCR analysis performed using a primers/probe sets targeting EBOV L-polymerase and *gapdh*. Data normalized to L-polymerase RNA levels on HEK293T LacZ CRISPR-TIM1 cells transfected with GFP based on ΔΔCt values. **(G)** EBOV trVLP normalized reporter activity on p2 target cells. HEK293T-TIM1 p1 cells were transfected with EBOV RNP proteins, and either GFP (grey) or ZAP-L (red) prior to infection with increasing amounts of wild-type EBOV trVLPs (trVLPwt, solid lines) or a variant with no CpG dinucleotides on the *Renilla* ORF of the 4cis genome (CpG low, dashed lines). Supernatants from p1 were harvested and used to infect HEK293T-TIM1 p2 target cells. **(H)** Quantification of viral RNA transcripts present intracellularly (upper and middle panels) and in supernatants (lower panel) of HEK293T-TIM1 cells transfected as in (G), and infected with a fixed amount of EBOV trVLP WT or CpG low. Random hexamer primers were used to generate cDNAs from total RNA and RT-qPCR analysis was performed as in (D). **(I)** EBOV trVLP normalized reporter activity on HEK293T-TIM1 p2 target cells. HEK293T LacZ CRISPR-TIM1 (grey) and HEK293T ZAP CRISPR KO-TIM1 cells (red) were transfected with EBOV RNP proteins and pre-treated with increasing amounts of IFN-I prior to infection with pre-determined and equivalent amount (1x10$^6$ RLU) of wild-type EBOV trVLPs (solid lines) or CpG Low EBOV trVLP (dashed lines). Supernatants from p1 were harvested and used to infect HEK293T-TIM1 p2 target cells. Reporter activities measured 24 hours post-infection. All the represented EBOV trVLP Renilla reporter activities are normalized to control Firefly luciferase values obtained in the same lysates. P > 0.05, **p > 0.01 and ***p > 0.001 as determined by two-tailed paired t-test. All error bars represent ± SEM of at least three independent experiments.

retroviruses, restriction of EBOV was not dependent on KHNYN, as depletion of this protein in U87-MG cells did not attenuate the antiviral effect of IFN-I on EBOV trVLPs replication nor rescue viral RNA levels (S2J and S2K Fig).

## TRIM25 interacts with the EBOV vRNP and promotes ubiquitination of NP

While the role of TRIM25 as a ZAP cofactor is established, how it facilitates ZAP targeting of viral RNAs is not understood. However, given the previously reported interaction between TRIM25 and influenza virus [53], we hypothesized that it could act directly on EBOV vRNP components to promote ZAP-mediated restriction. We noted that overexpression of TRIM25 induced a smear of higher molecular weight species above NP in western blots of transfected HEK293T cell lysates during the screen (S1D Fig), so we therefore first examined whether TRIM25 could interact with the EBOV NP and its chaperone VP35, known to link NP to L in order to facilitate transcription and replication of EBOV genome. NP co-immunoprecipitated with TRIM25, whereas VP35 only pulled-down with TRIM25 in the presence of NP (Fig 5A, left panel). Furthermore, reciprocal immunoprecipitation of NP from p1 cells brought down both VP35 and TRIM25, suggesting initially that TRIM25 could only interact with the vRNP in an NP-dependent manner (S3A Fig). This interaction was resistant to both RNase A and RNase III treatments, suggesting that the vRNA was not bridging the interaction between TRIM25 and NP (S3B Fig). However, in a reverse pull-down for VP35 we were able to show that TRIM25 can interact with VP35 in a NP-independent fashion, although this interaction was stronger in the presence of NP, together highlighting the capacity of this protein interacting with the vRNP (Fig 5A, right panel). Endogenous ZAP-L, but not ZAP-S, could also be detected in TRIM25/NP coprecipitates. However, in cells lacking ZAP, TRIM25 could still interact with NP (S3C Fig). To determine if TRIM25 localized to viral replication compartments, we analysed endogenous TRIM25 expression by immunofluorescence. TRIM25 re-

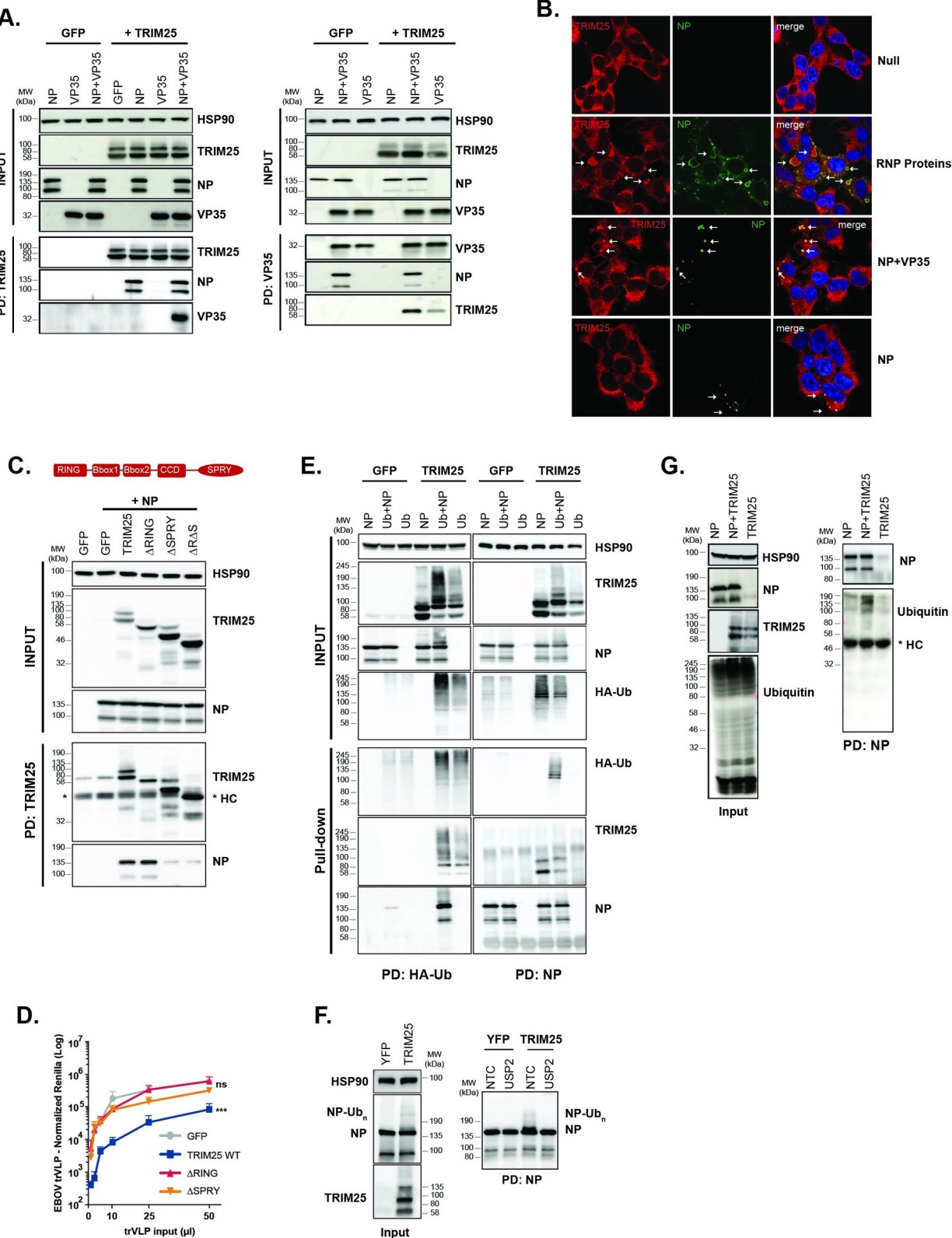

**Fig 5. TRIM25 interacts with EBOV NP and promotes its ubiquitination. (A)** Lysates of HEK293T-TIM1 cells transfected either with GFP or TRIM25, in combination with EBOV NP and/or EBOV VP35, were immunoprecipitated with an anti-TRIM25 (left panel) or an anti-VP35 (right panel) antibodies. Cellular lysates and pull-downs were analysed by western blot for HSP90, TRIM25, EBOV NP and VP35. **(B)** Panels show

representative fields for the localization of EBOV NP and endogenous TRIM25 on HEK293T-TIM1 cells left untreated (Null), or transfected with EBOV NP protein alone, or in combination either with VP35 or all remaining RNP proteins (VP35, VP30 and L). Cells were stained 24 hours post-transfection with anti-TRIM25 (red) and anti-NP (green) antibodies, as well as with DAPI (blue). White arrows point to the localization of TRIM25 intracellular aggregates. **(C)** Schematic representation of functional domains within TRIM25 (upper panel). Lysates of HEK293T-TIM1 cells transfected with EBOV NP in combination with GFP, TRIM25 or mutants thereof, were immunoprecipitated with an anti-TRIM25 antibody. Input and pull-down samples were blotted for HSP90, TRIM25 and EBOV NP (lower panel). (*) indicates the detected heavy-chains (HC) from the antibody used in the pull-down. **(D)** EBOV trVLP normalized reporter activity on HEK293T-TIM1 cells transfected with EBOV RNP proteins in combination with GFP (grey), TRIM25 wild-type (blue), TRIM25 ΔRING (red) or TRIM25 ΔSPRY (orange) mutants, prior to infection with increasing amounts of EBOV trVLPs (p1 target cells). EBOV trVLP Rluc reporter activities were measure 24 hours post-infection and normalized to control Fluc values obtained in the same lysates. EBOV trVLP Renilla reporter activities are normalized to control Firefly luciferase values obtained in the same lysates. *p > 0.05, **p > 0.01 and ***p > 0.001 as determined by two-tailed paired t-test. All error bars represent ± SEM of at least three independent experiments. **(E)** HEK293T-TIM1 cells were transfected either with GFP or TRIM25, in combination with EBOV NP and/or a plasmid expressing a HA-tagged Ubiquitin (HA-Ub). Lysates from these cells were immunoprecipitated with an anti-HA antibody (left panels) or an anti-NP antibody (right panels). Cellular lysates and pull-down samples were analysed by western blot for HSP90, TRIM25, EBOV NP and HA (ubiquitin). **(F)** Lysates from HEK293T cells co-transfected with EBOV NP and YFP or TRIM25 were immunoprecipitated with an anti-NP antibody, and pulled-down fractions treated with USP2 deubiquitinase enzyme. Cellular lysates and pull-downs were analysed by western blot for HSP90, TRIM25 and EBOV NP. **(G)** HEK293T-TIM1 were transfected with EBOV NP and/or TRIM25 under endogenous Ubiquitin levels and treated with Bafilomycin A (100nM). Lysates from the cells were immunoprecipitated with an anti-EBOV antibody. Cellular lysates and pull-down samples were analysed by western blot for HSP90, TRIM25, EBOV NP and Ubiquitin.

localized to NP-containing viral inclusion bodies in cells transfected with the vRNP proteins (Fig 5B). NP expression was sufficient for this re-localization, but it was markedly enhanced in the presence of VP35 which is known to regulate the conformational dynamics of NP in the vRNP [22].

Like many TRIM family members, TRIM25 consists of a N-terminal RING domain containing E3-ubiquitin ligase activity, two B-Box domains, a coiled-coil domain that allows it to form antiparallel dimers, and a C-terminal PRYSPRY substrate recognition domain [50,72,73] (Fig 5C). Using deletion mutants of TRIM25, we found that the mutants lacking the PRYSPRY domain, but not the RING domain, considerably lost much of their ability to interact with NP in co-immunoprecipitations (Fig 5C). However, both PRYSPRY and RING domains were essential for antiviral activity against EBOV trVLPs when overexpressed in HEK293T-TIM1 p1 cells (Fig 5D). When activated by their cognate ligands, many TRIM proteins undergo auto-ubiquitination as well as inducing ubiquitination of target proteins [74]. Consistent with this, in the presence of HA-tagged ubiquitin, co-expression of TRIM25 and NP induced the appearance of higher molecular weight species of both proteins, that could be precipitated with anti-HA and NP antibodies, indicating the presence of ubiquitin molecules (Fig 5E). Ubiquitinated species of NP and TRIM25 could not be detected in cells transfected with the RING-deleted TRIM25 mutant, indicating the RING E3 ligase activity was essential (S3D Fig). However, in HEK293T cells lacking either ZAP or FL-MAVS and miniMAVS where TRIM25 antiviral activity is curtailed, TRIM25 and NP co-expression could still lead to ubiquitination of both proteins (S3E and S3F Fig). Importantly, residual levels of NP ubiquitination observed in GFP transfected cells (Figs 5E, and S3D–S3F), are not detectable on HEK293T TRIM25 KO cells (S3G Fig). To exclude the possibility that our observations are product of an artefact associated with overexpressing HA-tagged ubiquitin, we pulled-down EBOV NP following its co-expression with YFP or TRIM25 under endogenous levels of ubiquitin and treated this fraction with Ubiquitin Specific Peptidase 2 (USP2). The typical higher molecular weight smear observed above NP when TRIM25 is ectopically expressed disappeared upon treatment with USP2, reinforcing our initial finding that TRIM25 promotes the ubiquitination of NP (Fig 5F). As TRIM25 is also an ISG15-ligase [75,76] and USP2 has been suggested to be cross-reactive and able to cleave both ubiquitin and ISG15 molecules from modified proteins, we further confirmed the ubiquitination status of pulled-down EBOV NP under endogenous levels of ubiquitin and expression of TRIM25. We observed an increased signal for endogenous ubiquitin on

immunoprecipitated samples at the expected molecular weight of EBOV NP following co-expression with TRIM25 in comparison with their single expression, while no evidence of ISGylated proteins was detectable in the same samples (Fig 5G). Of note, we did not detect evidence of a differential NP degradation in the absence of TRIM25 (S3H Fig). Thus, the ubiquitin ligase activity of TRIM25, and specifically the ubiquitination of NP and auto-ubiquitination of TRIM25, are necessary but not sufficient to explain its antiviral activity against EBOV.

Recent evidence has suggested that activation of TRIM25 E3 ligase activity requires its binding to RNA, with two distinct RNA interaction domains being described–a basic patch with 7 lysine residues in the linker between the coiled-coil and the PRYSPRY domain, and an RNA-binding domain (RBD) in the N-terminal region of the PRYSPRY itself [77,78]. Activation is also strictly dependent on the multimerization of TRIM25 dimers through dimerization of N-terminal RING domains [78]. In order to further probe the determinants of TRIM25-mediated antiviral activity, we made a series of mutations in these functional domains (Fig 6A). Compared to the wildtype protein, some of the targeted mutations in the RING dimerization interface, the catalytic site itself, or the PRYSPRY domain significantly impaired antiviral activity (Fig 6A). Of these RING dimerization mutants N66A, L69A and V72A lost all antiviral activity (Fig 6A). Similarly, deletion of the RBD or mutation of the linker region basic patch (7KA) led to lost activity. Targeted mutations predicted to disrupt the coiled-coil [73] were not significantly defective in restricting trVLP replication. Western blot analyses indicated that antiviral potency appeared to correlate well with NP ubiquitination under endogenous levels of ubiquitin (Fig 6B). While most RING dimerization mutants blocked ubiquitination of NP, individual mutations in the catalytic site reduced, but did not completely, abolish it. Furthermore, the effectively antiviral coiled-coil mutants Y254A and Y252A, together with the BBox2 RHK mutant, were still capable to induce NP ubiquitination. Interestingly the putative RNA-binding activity mutants, particularly the RBD, were also impaired in NP ubiquitination. All the functional RING mutants of TRIM25 tested retained the ability to interact with NP in coimmunoprecipitations as did the RBD deletion and unlike the full deletion of the PRYSPRY domain that is defective for this interaction (Fig 6C). Surprisingly, the 7K mutant lost most of NP interaction, suggesting that it may affect PRYSPRY-mediated NP interaction.

## TRIM25 is recruited to incoming vRNPs and reduces NP association with trVLP vRNA and promotes its interaction with ZAP

The data presented above support the hypothesis that TRIM25 and ZAP target the EBOV vRNP to block its transcription and/or replication. To examine this in detail we first tested whether incoming vRNPs alone were capable of re-localizing endogenous TRIM25. HEK293T LacZ CRISPR-TIM1 cells were infected with concentrated trVLP supernatants in the absence of transfection of vRNP components, then fixed 4 to 6 hours post-infection and stained for NP and TRIM25. Approximately 5% of cells were visibly infected, and in all of them distinct NP positive foci were detected in the cytoplasm, associated with an aggregation of endogenous TRIM25 (Fig 7A, left panel). These NP positive foci, given their size, suggest that the signals observed represent accumulations of several rather than single vRNPs. To ensure this re-localization was specific for cells in which trVLPs had entered, we made use of HEK293T-TIM1 cells in which the endosomal EBOV fusion receptor NPC1 had been deleted by CRISPR, thus rendering them uninfectable by VLPs pseudotyped with EBOV GP (S4A–S4C Fig). By contrast, in these cells we saw no evidence of cytoplasmic NP structures or TRIM25 re-localization (Fig 7A, right panel), suggesting that TRIM25 can only associate with incoming EBOV vRNPs after fusion of the viral membrane and release of the vRNPs into the cytoplasm.

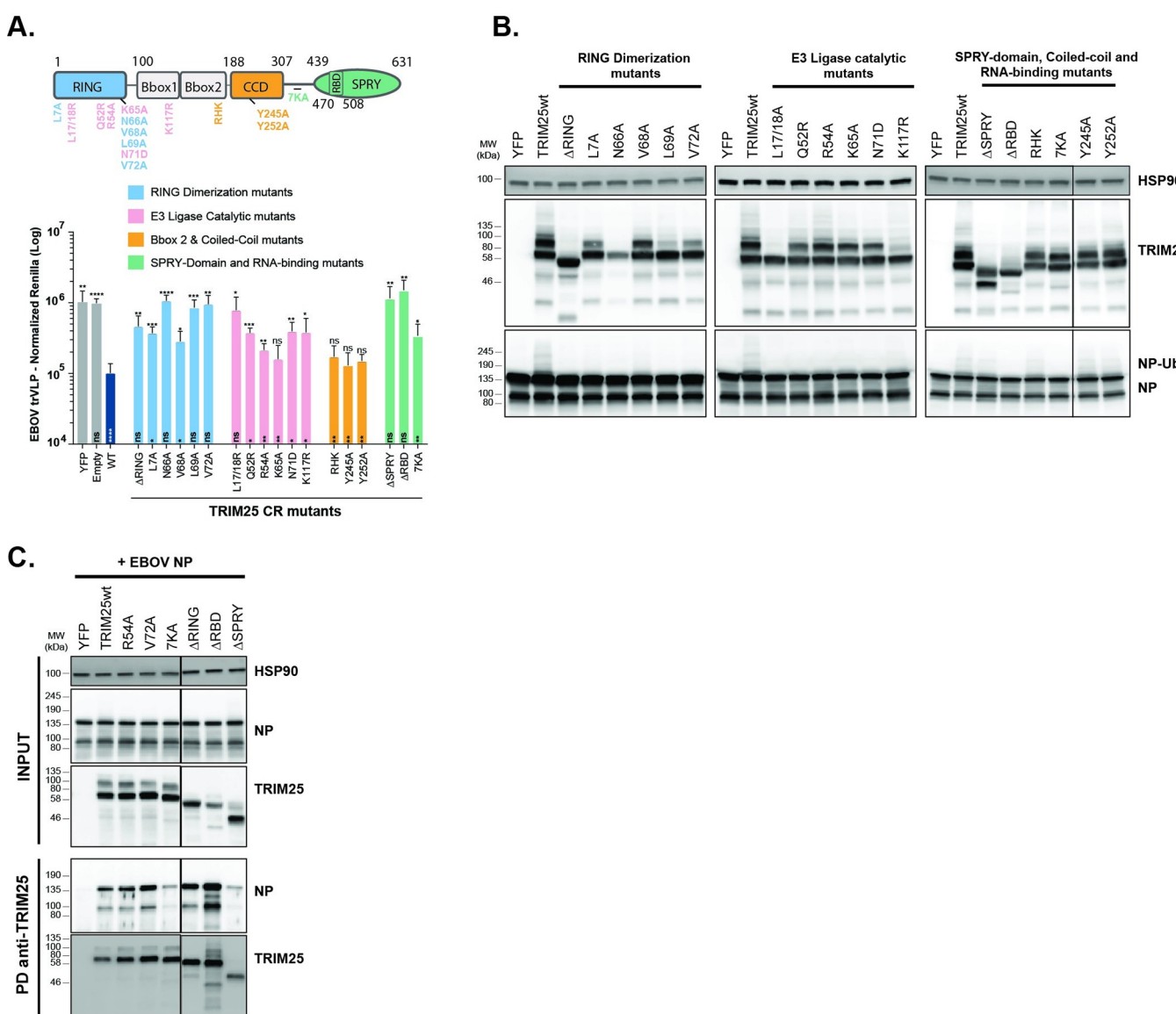

**Fig 6. Determinants of TRIM25 antiviral activity and NP interaction. (A)** Upper Panel: schematic representation of TRIM25 with the localization of the RING dimerization mutants (light blue), E3-Ligase catalytic mutants (pink), Bbox2 & coiled-coil mutants (orange) and SPRY-domain and RNA-binding mutants (light green) generated on a TRIM25 CRISPR-resistant background. Lower Panel: EBOV trVLP normalized reporter activity on HEK293T TRIM25 CRISPR KO-TIM1 cells transfected with EBOV RNP proteins in combination with GFP (grey), CRISPR-resistant (CR) TRIM25 wild-type (dark blue), or mutants thereof (see upper panel), prior to infection with EBOV trVLPs (p1 target cells). EBOV trVLP Rluc reporter activities were measure 24 hours post-infection and normalized to control Fluc values obtained in the same lysates. All error bars represent ± SEM of four independent experiments. $^*p > 0.05$, $^{**}p > 0.01$ and $^{***}p > 0.001$ as determined by One-way ANOVA. Statistics represented above graphic bars were calculated as multiple comparisons to TRIM25 wild-type, while the statistics within graphic bars are represented in function of multiple comparison to YFP. **(B)** HEK293T TRIM25 CRISPR KO cells were transfected with plasmids expressing the EBOV RNP proteins together with YFP, CRISPR-resistant TRIM25 or mutants thereof. Cell lysates were analysed 48 hours later by western blot for the expression of HSP90, TRIM25 and EBOV NP. **(C)** Lysates of HEK293T cells transfected with EBOV NP in combination with either YFP or CRISPR-resistant TRIM25 (or mutants thereof) were immunoprecipitated with a rabbit anti-TRIM25 antibody. Cellular lysates and pull-down samples were analysed by western blot for HSP90, TRIM25 and EBOV NP.

We then asked whether TRIM25 or ZAP could affect the integrity of the vRNP. As a control we could show that neither overexpression of TRIM25, nor ZAP, affected total cell-associated trVLP genomes 3 hours post-infection (S4D Fig). mRNA abundance is very low at this time-point, showing that any primary transcription is below the limit of detection. Thus, under

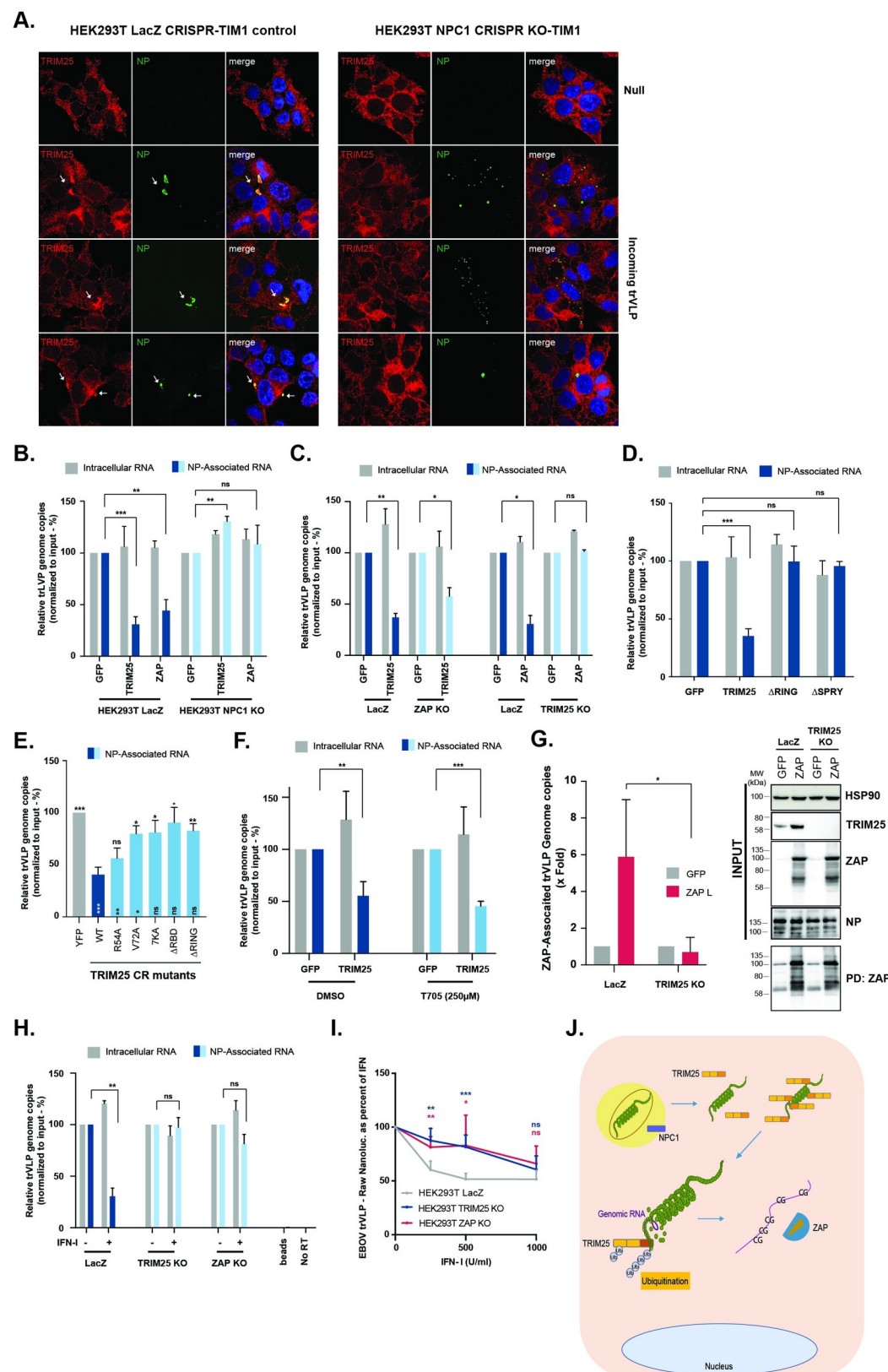

**Fig 7. TRIM25 and ZAP promote the dissociation of EBOV trVLP genomic RNA from the viral ribonucleoprotein.**
**(A)** Typical confocal microscopy fields from HEK293T LacZ CRISPR-TIM1 (left panels) or HEK293T NPC1 CRISPR

KO-TIM1 cells (right panels) left untreated (Null) or infected with EBOV trVLPs concentrated on a 20% sucrose-cushion. Cells were stained 4 to 6 hours post-infection with anti-TRIM25 (red), anti-EBOV NP (green) and DAPI (blue). White arrows show localization of TRIM25 intracellular aggregates. **(B)** Relative quantification of intracellular RNA levels (grey) and NP-associated RNA (blue) on HEK293T LacZ CRISPR-TIM1 and HEK293T NPC1 CRISPR KO-TIM1 cells transfected with GFP, TRIM25 or ZAP-L prior to infection with EBOV trVLPs. 3 hours post-infection cells were UV cross-linked, and EBOV NP from incoming virions were immunoprecipitated from lysates with an anti-NP antibody. Following proteinase K treatment, pulled-down RNA was extracted with Qiazol / chloroform, and random hexamer primers were used to generate cDNAs, and qPCR analysis performed using a primers/probe set targeting EBOV VP40 RNA. Values are presented as percentage of absolute RNA copy numbers on cells transfected with GFP. **(C)** HEK293T LacZ CRISPR, HEK293T ZAP CRISPR KO and HEK293T TRIM25 CRISPR KO cells stably expressing TIM1 were transfected with GFP, TRIM25 or ZAP-L as depicted in the panels, and later infected with EBOV trVLPs. Relative quantification of intracellular viral RNA levels (grey) and NP-associated RNA (coloured bars) were determined as in (B). **(D)** HEK293T-TIM1 cells were transfected with GFP, wild-type TRIM25 or mutants thereof prior to infection with EBOV trVLPs. Relative quantification of intracellular viral RNA levels (grey) and NP-associated RNA (blue) were determined as in (B). **(E)** HEK293T TRIM25 CRISPR KO-TIM1 cells were transfected with GFP (grey), or CRISPR-resistant versions of TRIM25 (wild-type, dark blue; or mutants thereof, as depicted in the figure, light blue) prior to infection with EBOV trVLPs. Relative quantification of NP-associated RNA was determined as in (B). **(F)** Prior to infection with EBOV trVLPs, HEK293T LacZ CRISPR-TIM1 cells were transfected with GFP or TRIM25 and either treated with 250μM of T705 (Favipiravir), or the equivalent volume of the diluent (DMSO). Relative quantification of intracellular viral RNA levels (grey) and NP-associated RNA (coloured bars) were determined as in (B). Values are presented as percentage of absolute RNA copy numbers on cells transfected with GFP. **(G)** Relative quantification of ZAP-associated RNA on HEK293T LacZ CRISPR-TIM1 and HEK293T TRIM25 CRISPR KO-TIM1 cells transfected with GFP (grey) or ZAP-L (red) prior to infection with EBOV trVLPs (left panel). 3 hours post-infection cells were UV cross-linked, and ZAP was immunoprecipitated from lysates. RNA extraction, cDNA synthesis and RT-qPCR analysis were performed as in (B). Values were normalized to the respective inputs and are presented relative to absolute RNA copy numbers on cells transfected with GFP. Cellular lysates and pull-down samples were analysed by western blot for HSP90, TRIM25, NP and ZAP (right panel). **(H)** HEK293T LacZ CRISPR, HEK293T TRIM25 CRISPR KO and HEK293T ZAP CRISPR KO cells stably expressing TIM1 were either untreated or pre-treated with 1000U/ml of IFN-I prior to infection with EBOV trVLPs. Relative quantification of intracellular viral RNA levels (grey) and NP-associated RNA (coloured bars) were determined as in (B). Values are presented as percentage of absolute RNA copy numbers on cells non-treated with IFN. **(I)** HEK293T LacZ CRISPR, HEK293T TRIM25 CRISPR and HEK293T ZAP CRISPR KO cells stably expressing TIM1 were pre-treated overnight with increasing concentrations of IFN-I prior to infection with EBOV nanoluciferase trVLPs. EBOV trVLP nanoluc reporter activities were measured 48 hours post-infection and data is shown as a percentage of untreated for each cell line individually. **(J).** Proposed model for the mechanism associated with the antiviral activities of TRIM25 and ZAP against EBOV trVLP. EBOV viral particle enters the cells by macropinocytosis, followed by NPC1-dependent fusion with the cellular endosomal membrane. Once in the cytoplasm, TRIM25 is recruited to the viral particle through an interaction with EBOV NP protein, leading to the ubiquitination of both viral target and TRIM25 itself. This results in the displacement of the viral RNA genome from the vRNP, followed by its recognition by ZAP in a way dependent of the genome's CpG content and subsequent impact in the transcription and replication of EBOV trVLP.

conditions in which no vRNP components were ectopically expressed, we performed RNA-IPs of NP-associated genomic trVLP vRNA after UV-crosslinking of cells overexpressing GFP, TRIM25 or ZAP. Immunoprecipitation of NP under these conditions consistently brought down 5-to-10% of the input genomes. Interestingly, overexpression of TRIM25 or ZAP significantly reduced the levels of trVLP genome associated with incoming NP (Figs 7B and S4E), independently of using WT or CpG low EBOV trVLPs (S4F Fig). Importantly, TRIM25 and ZAP did not dissociate EBOV trVLP RNA from NP in NPC1 CRISPR knockout cells where cell-associated virions will remain unfused in endosomal compartments (Fig 7B). Furthermore, whilst TRIM25 overexpression in ZAP KO cells could still reduce NP/genome association, ZAP overexpression could not mediate the same effect in TRIM25 KO cells (Fig 7C), indicating that TRIM25, not ZAP, was essential for this effect, while the CpG content of the viral genome does not influence the targeting of the RNP by TRIM25 (S4G Fig). Furthermore, neither RING nor PRYSPRY-deletion mutants of TRIM25 could promote NP/genome dissociation (Figs 7D and S4H), neither could the RNA-binding or RING mutants (Figs 7E and S4I), consistent with their lack of antiviral activity.

Since transcription of viral genes by the vRNP-associated RdRp is one of the first events in EBOV replication after cellular entry, we performed the experiment in Fig 7B in the presence of 250μM of the L inhibitor T705, which is sufficient to inhibit trVLP RLuc expression by 90%

(S4J and S4K Fig). Importantly, we found that T705 did not prevent TRIM25 overexpression from dissociating NP from the genome of incoming viruses (Fig 7F). Together these data indicate vRNP interaction, RNA-binding and activation of E3 ligase activity are required for TRIM25 to dissociate NP from the viral genome of incoming EBOV vRNPs, and thus facilitate interaction with ZAP.

We then performed a similar crosslinking-IP in cells, this time precipitating overexpressed ZAP. This markedly enriched for the trVLP viral genome at the same timepoint that NP/genome association was reduced (Fig 7G). However, immunoprecipitation from TRIM25 KO cells under the same experimental conditions showed no evidence of ZAP association with the trVLP genome, indicating that this correlated with TRIM25-mediated NP/genome dissociation. To further address the importance of the CpG content for ZAP association, we used a monocistronic genome containing the *Rluc* gene flanked by the leader and trailer regions of EBOV, from which we generated a low CpG variant in which all the CpGs in *Rluc* were silently mutated (S4L Fig). As expected, ZAP overexpression impacts on the replication of the WT monocistronic genome, whilst its effect is attenuated upon reduction of the CpG content (S4M Fig). This correlated with the significant impairment of ZAP association with the CpG low monocistronic genome when compared with the wild-type (S4N Fig), thus highlighting the important role of the viral genome CpG content for ZAP-mediated antiviral effect.

Finally, we asked if the NP/genome dissociation could be recapitulated following an IFN-I treatment. HEK293T LacZ-TIM1 cells, or those lacking TRIM25 or ZAP were treated overnight with IFN-I and then infected for 3h with trVLPs and similarly subjected to UV-crosslinked RNA-IP of NP. In HEK293T LacZ-TIM1 cells, IFN-I pre-treatment reduced the NP/genome association to a similar extent to the ectopic overexpression of TRIM25, without affecting total cell-associated trVLP vRNA. However, in TRIM25 knockout cells, there was no significant reduction in NP/genome association. In ZAP knockout cells, whilst NP/genome interactions were largely restored, there remained a small reduction in trVLP vRNA recovered after NP immunoprecipitation (Fig 7H). Thus, the TRIM25 dependent dissociation of NP from the trVLP genome can be demonstrated in cells after IFN-I stimulation, implying the relevance of these observations to the antiviral mechanism.

In order to further investigate whether we could observe the antiviral effect of TRIM25 and ZAP on an alternative model of the trVLP system based only on transduction, we created an EBOV minigenome that expressed a nanoluciferase in place of the Renillla, resulting in a more sensitive reporter capable of reliably detecting at low enough quantities to observe primary transcription without the requirement of transfecting vRNP components. The observed primary transcription could be quantified specifically as luciferase signal was not observed with either ΔGP trVLPs or in NPC1 CRISPR knockout cells (S4O Fig). We then investigated the effect of TRIM25 and ZAP and observed that the antiviral activity of IFN-I treatment on primary transcription was attenuated in either TRIM25 or ZAP knockout cells at lower concentrations (Fig 7I). Furthermore, we also observed that TRIM25 overexpression was also antiviral in this system (S4P Fig). These observations indicate that both TRIM25 and ZAP are important for the cellular defence against EBOV in a transduction-only based model.

## Discussion

Through screening of a human ISG library we provide evidence that TRIM25 and ZAP are major effectors of interferon-induced antiviral restriction of EBOV trVLP. The data presented herein lead us to propose the following model for the mechanism of this restriction (Fig 7J). TRIM25 interacts with the viral NP and is recruited to the incoming vRNP shortly after fusion of the virion with the cellular endosomal membrane. This results in the ubiquitination of both

NP and TRIM25, and the apparent displacement of NP from the viral genome, facilitating ZAP binding to the vRNA. The antiviral inhibition of the subsequent transcription and replication of the virus is dependent on this interaction, and the overall sensitivity of the virus to both ZAP and IFN-I can be modulated by the CpG content of the viral genomic RNA. This activity of TRIM25 is completely independent of RIG-I-mediated sensing of viral RNA and proinflammatory signalling but requires both isoforms of MAVS. As such we propose this as a direct antiviral restriction mechanism associated with, but independent of, classical cytoplasmic RNA pattern recognition.

Until recently, TRIM25 was thought to be the essential cofactor for the RIG-I helicase to recognise viral RNA species, particularly those with exposed 5' triphosphates [50]. However, it was suggested that the E3 ligase RIPLET acts with TRIM25 in a sequential manner to activate RIG-I upon viral RNA engagement [60], whereas other data implicated RIPLET as being sufficient to ubiquitinate and activate RIG-I [59]. Thus, the true contribution of TRIM25-mediated ubiquitination of the RIG-I 2-CARD motif to the downstream signalling is still debatable. Upon activation, RIG-I and MAVS undergo a prion-like polymerization into extended filaments that directly activate the kinase TBK1, and thus promote proinflammatory signalling [79], but it is also still unclear whether TRIM25 plays further roles in the pathway. However, although we cannot formally rule out an involvement of later components of the IFN-mediated pathway, such as the activation of other ISGs, neither RIG-I nor TBK1 are necessary for TRIM25 to inhibit EBOV trVLP replication, implying a role distinct from classical RNA sensing.

The PRYSPRY domain of TRIM25 binds to and activates ZAP [51,54], which has long been known as an antiviral effector targeting various RNA viruses [56,63,64]. For a number of positive sense RNA viruses, ZAP binds to the RNA genome and restricts replication by blocking its translation and/or targets viral RNA for degradation by the 3'-5' exosome [51,54,63]. ZAP exists as two isoforms, ZAP-L and ZAP-S, the latter being a splice variant lacking the C-terminal catalytically inactive poly-ADP ribosyl polymerase (PARP) domain [65]. Whilst antiviral activity has been ascribed to both isoforms, ZAP-L is more active against some viruses [51,54,63,64]. Upon overexpression, ZAP-L restricts EBOV trVLP more potently than ZAP-S, and we find that only ZAP-L coprecipitates with TRIM25 and NP. Although the limitations associated with the use of the EBOV trVLP system, and the fact we didn't validate our restriction phenotypes on experiments with full-length EBOV, it was previously shown that overexpression of a rat ZAP has an antiviral activity against full-length EBOV by blocking the translation of L mRNA [69]. While we also see an effect on both L mRNA and protein expression upon ZAP overexpression in our assays, this is also observable in TRIM25 KO cells where ZAP has no antiviral activity, in contrast to the impact of ZAP overexpression on EBOV genomic RNA levels which occurs in a TRIM25-dependent manner. Moreover, the TRIM25/ZAP-mediated block induced by IFN-I does not affect the expression of any viral protein expressed *in trans* and ZAP activity is modulated by the CpG content in Renilla, suggesting that ZAP is targeting the viral genome rather than just blocking the synthesis of L.

ZAP has been implicated in driving the evolution of RNA viruses towards suppression of their genomic CpG content [56,70]. Most RNA viruses have a lower abundance of CpGs than would be expected by chance, and in the case of HIV-1, artificially increasing the CpG content in parts of the genome renders the virus sensitive to ZAP [56,71,80]. Evidence suggests ZAP directly binds to viral RNAs with high CpG content and promotes their degradation [55,56]. EBOV exhibits less CpG suppression than HIV (observed vs expected ratio of 0.5 compared to 0.2). However, if this CpG content in either the trVLP or monocistronic systems is changed, the viral sensitivity to ZAP is modulated, and in the case of the latter, ZAP/RNA interactions enriched. Since *RLuc* expression is not required for the trVLP propagation, this result implies that ZAP can target CpGs in the EBOV genome itself.

How TRIM25 activates ZAP remains unclear. TRIM25 interacts with and ubiquitinates ZAP, but the latter activity is not required for its cofactor activity [51,54]. Our data showing that TRIM25 displaces NP from the genome shortly after entry and that ZAP binds the vRNA in a TRIM25-dependent manner raises the possibility that its role is to expose viral RNA rather than acting directly on ZAP itself. Importantly, through this TRIM25/ZAP axis, we show that the CpGs in the genome render EBOV sensitive to IFN-I. While both TRIM25 and ZAP are IFN-inducible, they are constitutively expressed in most cells and only increase modestly at the protein level upon IFN stimulation. Therefore, despite being essential effectors targeting EBOV trVLP replication, whether this increase in their expression fully explains the IFN-regulated antiviral activity, or whether other IFN-inducible components are required needs further study [43].

Our results indicate that TRIM25 may target the structure of the EBOV (and perhaps other negative strand RNA virus) vRNP in a manner analogous to antiviral restriction by other mammalian proteins. While we see no evidence of TRIM25-dependent NP degradation, TRIM25 does appear to displace NP from the viral RNA, which would likely dysregulate viral transcription and RNA replication. Similar to rhesus TRIM5 and HIV CA, as well as TRIM21 and its antibody opsonised targets [81,82], we see evidence of NP ubiquitination. While this correlates with antiviral activity, we do not yet know whether that is itself essential. Also, we have not demonstrated that TRIM25 directly ubiquitinates the NP protein, merely that TRIM25 overexpression results in the ubiquitination of NP. Furthermore, although we have not seen evidence for the occurrence ISGylation under our experimental conditions, we cannot exclude an eventual role for TRIM25-mediated ISGylation of NP under conditions where ISG15 is present and/or induced.

TRIM25 forms an antiparallel dimer through its coiled-coil domain [73] and can then further assemble into multimers through RING domain dimerization. This positions the RING moieties at either end of the dimer, with the ligand-binding PRYSPRY located more centrally, either side of the coiled-coil. To act as an E3 ligase the TRIM25 RING domain must also dimerize, which can only happen through association with another TRIM25 dimer. This has been demonstrated to be facilitated by the binding of the PRYSPRY domain to one of its cognate ligands, the RIG-I 2CARD, generating the catalytically active domain that can interact with E2 ligases [83]. Thus, TRIM25 is only active as a multimer of dimers when bound to its target. Two possibilities exist for the conformation of these multimers from the current structural studies of TRIM25; either two TRIM25 dimers multimerize into a closed dimer of dimers, or end-to-end multimerization of RING domains could lead to extended polymers of TRIM25 [83]. We find evidence that TRIM25 interaction with NP depends on its PRYSPRY domain. Antiviral activity, however, was blocked by mutations that impair RING dimerization as well as catalytic activity, suggesting that multimerization of TRIM25 on the EBOV vRNP is also essential. Since this is a helical ribonucleoprotein complex, it is possible that end-to-end multimers of TRIM25 could form over such a structure, driving its disassembly, and thus recruit other RNA-directed antiviral factors and sensors.

Several studies have suggested that TRIM25 is an RNA-binding protein itself and that this is important for the activation of the E3 ligase activity. Two RNA binding sites have been identified: (1) A tyrosine-rich segment of loops 2 and 3 of the PRYSPRY domain has been shown to bind the Let7 miRNA both *in vitro* and *in cellulo*, and CLIP analyses has revealed many cellular TRIM25-associated RNAs with a preference for G-rich regions [77]. (2) A second domain, a basic patch in the unstructured region that links the coiled coil to the PRYSPRY, had also been shown to bind RNA *in vitro* [78]. In both cases, RNA-binding has been shown to promote E3 ligase activity *in vitro*, and mutation of these sites has been shown to block the substrate ubiquitination. It is unknown whether both domains are functionally independent,

or whether their mutation have effects other than RNA-binding on TRIM25 substrate recognition. Interestingly, in the case of EBOV restriction we find distinct phenotypes for either site. Mutation of the PRYSPRY RBD loses antiviral activity and NP ubiquitination but retains NP interaction consistent with this being RNase insensitive. Mutation of the basic patch (7KA) similarly loses antiviral activity, but unlike the RBD deletion, also loses most of the NP interaction similar to a full PRYSPRY deletion. Given that the basic patch lies within a linker between the coiled-coil and the PRYSPRY, it is possible that such a change may affect the spacing and positioning of the PRYSPRY relative to the coiled-coil for multimeric substrate engagement, as is well-known for other TRIM-family members. The implication of RNA-binding in TRIM25's antiviral mechanism is attractive in the light of our data that shows that it can dissociate NP from the genome even when transcription is inhibited. Cryo-EM structures of the EBOV vRNP show RNA tightly wound around helical NP assemblies, with each protomer engaging 6bp of RNA [84,85]. The RNA is clamped into a cleft, reducing its accessibility to cellular factors. In such a configuration, TRIM25 would have to interact with NP to access the vRNA, which is consistent with our mutagenesis data. Recent studies have shown that nuclear TRIM25 associates with influenza virus vRNPs to block transcriptional elongation [53]. While there are differences in these studies taken together, these observations raise the possibility that TRIM25 may have a broad direct antiviral activity against negative-strand RNA viruses with helical vRNPs. More widely, these data also suggest that TRIM25 can act as a ZAP cofactor by disassembling RNA/protein structures to expose CpG-rich RNAs.

Further to their direct antiviral restriction, several TRIMs ligand-induced synthesis of K63 polyubiquitin chains act as platforms for the activation of TAK1 and subsequently NFkB-dependent proinflammatory gene expression [81,86]. TRIM5 acts effectively as a pattern recognition receptor for the hexameric array of retroviral capsids, and TRIM21 for antibody-bound microbes entering the cytoplasm. This therefore raises the question of whether the "classical" cytoplasmic sensing mechanisms themselves are linked to a direct virus restriction. The requirement for MAVS in TRIM25/ZAP-mediated restriction of EBOV replication would suggest that this may be the case for the cytoplasmic RNA sensing pathway. Intriguingly, we find that both isoforms of MAVS are required, including miniMAVS which is suggested to be a negative regulator of RNA sensing [62]. Since depletion of TBK1 had no impact on TRIM25-mediated restriction, these observations together suggest that signalling through MAVS is not required for the direct antiviral activity. Thus, we suggest that MAVS may act as a regulatory part of the complex. MAVS localizes primarily to mitochondrial associated membranes and peroxisomes, and as such forms a platform for RNA sensing via RIG-I and MDA5. It is becoming clear that MAVS-dependent antiviral responses are of primordial importance, and that are controlled by protein catabolic processes as the ubiquitin-proteosome system and autophagy [87], in order to avoid inflammation and cell death. We suggest that these MAVS-induced platforms may also allow the recruitment of directly acting antiviral factors, as well as a link to the ultimate disposal of viral components by the cell. For example, various studies have linked MAVS, and its functional paralogue in DNA sensing, STING, to autophagic degradation upon activation [88,89].

TRIM25 is a common target for antagonism by RNA viruses. Influenza A virus NS1 blocks TRIM25 activity to disable RIG-I in infected cells [90], whereas dengue virus encodes a subgenomic RNA that functions analogously [91]. However, the interpretation of these findings might be revisited given the newly defined essential role of the other E3 ligase RIPLET on RIG-I activation [59]. We see no evidence of a direct counteractivity for human TRIM25 encoded by EBOV. VP24, which blocks signalling from IFNAR1/2, prevents the upregulation of ISGs after IFN-I exposure. In cells already exposed to IFN-I at the time of infection, VP24 would not be expected to have any effect, and in any case is encoded by the trVLP. VP35, is

expressed *in trans* in the trVLP system and counteracts RIG-I and protein kinase R (PKR) but has no known activity to inhibit TRIM25 [4]. However, humans are not the virus' natural host, and it is possible that filoviruses do have a countermeasure/resistance mechanism to TRIM25/ZAP in other mammals, particularly putative bat reservoir species [8,9]. Interestingly, adaptation of EBOV to lethal infection of both mice and guinea pigs is associated with amino acid changes in NP. Furthermore, lethality of mouse adapted EBOV in C57BL/6 mice requires the knockout of *mavs* [92]. Whilst this has been interpreted to be due to enhanced RNA sensing, given our observations that both MAVS isoforms are required for TRIM25/ZAP-mediated antiviral activity, it is also possible that differences in direct restriction of the vRNP underlie this species-specific tropism. Both ZAP and TRIM25 are under high levels of positive selection in mammals, suggesting a continuous adaptation to new viral pathogens [65].

In summary, our studies have revealed that TRIM25 and ZAP are major contributors to an IFN-induced restriction of EBOV replication. We provide evidence that TRIM25 directly associates with the incoming vRNP and dissociates NP from the genome. These data further suggest TRIM25 as a key restriction factor for EBOV, and that its role is to expose CpG rich areas of the viral genome to ZAP-mediated antiviral activities.

## Material and methods

### Cell lines

The parental cell lines HEK293T (CRL-3216) and U87-MG (HTB-14) were obtained from ATCC. HEK293T-TIM1 and U87-MG-TIM1 cells were produced by transduction of the parental cell lines with an MLV-based lentiviral vector packaging a pCMS28 vector genome [93] encoding the TIM1 construct, and selecting the cells with puromycin. Alternatively, some of the CRISPR KO cell lines (described below) were derived to stably express TIM1 by transduction with an MLV-based retroviral vector packaging a pLHCX (Clontech) vector genome encoding TIM1 and selecting with hygromycin. All cells used in this study were maintained in high glucose DMEM supplemented with GlutaMAX, 10% fetal bovine serum (FBS), 20μg/mL gentamicin, and incubated at 37˚C with 5% $CO_2$.

### Generation of CRISPR Knock-Out cell lines

CRISPR guides targeting *E.coli lacZ* gene, or human TRIM25, ZAP, RIG-I, FL-MAVS, FL-MAVS/miniMAVS, TBK1 and KHNYN genes were cloned into *BsmBI* restriction enzyme sites in the lentiviral CRISPR plasmid lentiCRISPRv2 (Addgene). LentiCRISPR VLPs were produced on HEK293T cells seeded on a 10cm dish, and transfected with 8μg of lentiCRISPRv2-Guide, 8μg of pCRV1-HIV-Gag Pol [94] and 4μg of pCMV-VSV-G [94]. Supernatants were harvested 48 hours later and used to transduce HEK293T or U87-MG cells, followed by selection in puromycin. Furthermore, MAVS isoforms expression were rescued on HEK293T FL-MAVS & miniMAVS CRISPR DKO by transduction with a MLV-based lentiviral vector packaging a pCMS28 vector genome encoding CRISR-resistant MAVS (MAVS^CR), miniMAVS (MAVS M1A^CR) or FL-MAVS (MAVS M142A^CR) constructs, and selecting cells with Blasticidin.

Additionally, the CRISPR guide targeting NPC1 gene was cloned into lentiCRISPR_v2_GFP as above (Addgene). The generated plasmid was transfected into HEK293T-TIM1 cells, which were subsequently sorted by flow cytometry, on a BD FACSCantoII, on the basis of their GFP expression. Finally, NPC1 expression was rescued on these HEK293T NPC1 CRISPR KO-TIM1 cells by transduction with an MLV-based lentiviral vector packaging a pCMS28 vector genome encoding NPC1 construct, and selecting the cells with Blasticidin to generate the HEK293T NPC1 CRISPR KO-TIM1 + NPC1 cell line.

All CRISPR guides sequences are in S2 Table.

## Generated plasmids

TRIM25wt, TRIM25 ΔRING and TRIM25 ΔSPRY were PCR amplified from the ISG library plasmid pcDNA-DEST40-TRIM25 [48] using forward and reverse primers encoding 5' overhangs with, respectively, *EcoRI* and *XhoI* restriction sites. Purified amplification products were then inserted into the corresponding sites on a pcDNA3.1 backbone. pcDNA3.1-TRIM25 ΔRINGΔSPRY was generated by PCR amplification using pcDNA3.1-TRIM25 ΔRING as template and sub-cloning the PCR product into pcDNA3.1 backbone as above.

pcDNA3.1-TRIM25wt CR (CRISPR-resistant) was generated by overlapping PCR from pcDNA3.1-TRIM25wt, using internal primers that inserted silent mutations on the CRISPR-guides target sequence and the above mentioned TRIM25 ORF flanking forward and reverse primers with *EcoRI* and *XhoI* restriction sites as 5' overhangs. pcDNA3.1-TRIM25wt CR was used as template for all CRISPR-resistant TRIM25 mutants used in this study. These mutants were generated, unless otherwise mentioned, by overlapping PCR using internal primers that inserted silent mutations, and the above mentioned forward and reverse primers encoding 5' overhangs with *EcoRI* and *XhoI* restriction sites. TRIM25 L7A CR was generated by PCR amplification of TRIM25wt CR with L7A_CR_Fwd primer with a 5' *EcoRI* overhang and the above mentioned TRIM25_*XhoI* Rev primers. Purified amplification products were then inserted into the corresponding sites on a pcDNA3.1 backbone.

pCMS28-Blast-MAVS<sup>CR</sup> was constructed by overlapping PCR from pCR3.1-MAVS [86], using internal primers that inserted a silent mutation in the NGG PAM of the CRISPR guide target sequence alongside forward and reverse primers encoding 5' overhangs with *NotI* and *XhoI* restriction sites respectively. pCMS28-Blast-MAVS<sup>CR</sup> was then used as a template and backbone for the MAVS CRISPR resistant plasmids used in this study. pCMS28-Blast-M142A<sup>CR</sup>, expressing only the FL-MAVS isoform, was created by overlapping PCR from pCMS28-Blast-MAVS<sup>CR</sup> using internal primers substituting the methionine for an alanine at position 142, in conjunction with the above mentioned forward and reverse primers with *NotI* and *XhoI* 5' overhangs. Furthermore, pCMS28-Blast-M1A<sup>CR</sup> was created by PCR amplification using forward and reverse primers encoding 5' overhangs with *NotI* and *XhoI* restriction sites respectively, where the forward primer also encoded for a mutation in the first methionine thus disrupting the expression of the FL-MAVS isoform. Amplified DNA fragments were then inserted into the pCMS28-Blast backbone digested with *NotI* and *XhoI*.

pCMS28-TIM1 was generated by sub-cloning the product of pCAGGS-TIM1 digestion with *EcoRI* and *NotI* enzymes into the corresponding sites in pCMS28 backbone. Additionally, pCAGGS-TIM1 served also as template for the PCR amplification of TIM1, and the product inserted into *XhoI* and *NotI* sites on pLHCX backbone to generate pLHCX-TIM1.

Finally, pBABE-NPC1 [16] served as template for the PCR amplification of NPC1, and the PCR product was digested with *EcoRI*/*XhoI* and inserted into the corresponding restriction sites in pCMS28-Blasticidin to generate pCMS28-Blast-NPC1.

All primers used for cloning are described on S3 Table.

To generate p4cis-vRNA-CpG low RLuc we have synthesised a 2141nt-long DNA fragment (sequence available upon request), with 61 silent mutations that removed all CpG dinucleotides present in the *Renilla* luciferase gene of the p4cis-vRNA-RLuc sequence. This DNA fragment was digested with *ApaI* and *BlpI* and inserted into the corresponding sites in the original p4cis-vRNA-RLuc minigenome plasmid. On the other hand, the monocistronic plasmid pT7-1cis-vRNA-CpG low-EBOV-hrluc expressing a Renilla luciferase with no CpG's was constructed by overlapping PCR using p4cis-vRNA-CpG low as template. Briefly, two fragments were amplified by PCR with the primers P4 cis Low CpG TRAIL F and P4 cis vRNA *XmaI* R (size 793bp) and P4 cis Low CpG TRAIL R and P4 cis vRNA *BlpI* F (size 1626bp). These DNA

fragments were joined together by PCR using the outer primers with the *BlpI* and *XmaI* restriction sites in their 5' overhangs, digested with these enzymes and cloned into the pT7-1cis-vRNA-EBOV-hrluc plasmid to create a low CpG Renilla monocistronic EBOV plasmid.

## Ebola transcription- and replication-competent virus-like particle (trVLP) assays

Unless otherwise stated, EBOV trVLPs [49] were produced in HEK293T producer cells (also called p0 cells) seeded in a 10 cm dish, and transfected with 625ng of pCAGGS-NP, 625ng of pCAGGS-VP35, 375ng of pCAGGS-VP30, 5μg of pCAGGS-L, 400ng of p4cis-vRNA-Rluc (or corresponding CpG low variant of this minigenome), and 250ng of pCAGGS-T7, using TransIT-LT1 transfection reagent. Medium was exchanged 4–6 hours post-transfection, and trVLP-containing supernatants were harvested 48–72 hours later and cleared by centrifugation for 5 minutes at 300xg. Titres of the different EBOV trVLP batches were then determined in HEK293T-TIM1 cells as RLU/ml. P1 target cells that were seeded on 24-well plates and transfected 24 hours before infection with 31.25ng of pCAGGS-NP, 31.25ng of pCAGGS-VP35, 18.75ng of pCAGGS-VP30, 250ng of pCAGGS-L (or pCAGGS-L-HA where mentioned) and 5ng of pFluc per well. In specific experiments, cells were also co-transfected at this stage with 125ng of the ISG of interest. Furthermore, if p1 target cells were not stably expressing TIM1, they were additionally co-transfected with 62.5ng of pCAGGS-TIM1 per well. Medium was changed 4–6 hours post-transfection and, in experiments performed either with interferons or T705 (Favipiravir), increasing amounts of universal IFN-I, IFN alpha 2a, IFN beta 1a (PBL Interferon Source) or T705 (AdooQ) were added to the cells at this stage. Typically, we added 0, 100, 250 and 500U/ml of universal IFN-I to the cells, unless specified otherwise, while using 0, 125 and 250U/ml of IFN alpha 2a or IFN beta 1a. Finally, we pre-treated cells with 0, 25, 50, 100, 200 or 250 μM of T705. Target p1 cells were infected the following day typically with a volume of EBOV trVLPs equivalent to $10^6$ RLU for HEK293T-based cell lines, and $10^5$ RLU for U87MG-based cells. Medium was again changed 4–6 hours after infection of p1 target cells, and 24 hours later cells were lysed with Passive Lysis 1x and reporter activities measured with Dual Luciferase Reporter Kit (Promega). At the same time supernatants from p1 cells were cleared by centrifugation and passaged onto fresh HEK293T-TIM1 p2 target cells, seeded in 24-well plates and transfected 24 hours earlier exclusively with the EBOV RNP components and pFluc (same quantities as above). These cells were lysed 24 hours post infection and reporter activities measured as already described.

Specific experiments required the use of concentrated EBOV trVLPs. For that purpose trVLPs were produced as above, and treated for 2 hours with 10U/ml DNase-I (Roche) before being concentrated by ultracentrifugation through a 20% sucrose/PBS cushion (28,000 rpm on a Sorvall Surespin rotor, for 90 min at 4˚C), and resuspended overnight in serum-free DMEM medium.

## ISG screen

ISG screen was conducted on HEK293T-TIM1 cells in a 96-well plate format, using a library of human ISGs encoded by pcDNA-DEST40 [48]. 50ng of individual ISG-expressing plasmids were co-transfected onto p1 target cells together with EBOV RNP expressing plasmids and pFluc, as described above in the trVLP assay. p1 target cells were infected with EBOV trVLPs 24 hours post-transfection and medium changed 4 hours later. 24 hours post-infection the cells were lysed with Passive Lysis 1x and reporter activities measured with Dual Luciferase Reporter Kit. At the same time supernatants from p1 cells were used to infect fresh HEK293T-TIM1 p2 target cells, seeded in 96-well plates and transfected 24 hours earlier exclusively with

the EBOV RNP components (as described above). These cells were lysed 24 hours post infection and reporter activities measured as already previously described. For each ISG, the yield of infectious virions was expressed as a percentage of the mean value across each library plate.

## Ebola monocistronic genome assays

VLPs carrying the monocistronic EBOV genomes were produced in HEK293T cells seeded in 10cm dishes, and transfected with 625ng of pCAGGS-NP, 625ng of pCAGGS-VP35, 375ng of pCAGGS-VP30, 300ng of pCAGGS-VP24, 5μg of pCAGGS-L, 1.25μg of pCAGGS-VP40, 1.25μg of pCAGGS GP, 1.25μg of pCAGGS-T7, 1.25μg of pT7-1cis-vRNA-EBOV-hrluc (or corresponding CpG low variant) and 125ng of pFLuc, using TransIT-LT1 transfection reagent. Medium was exchanged 4–6 hours post-transfection and supernatants harvested 72 hours later as described before. Experiments with these monocistronic VLPs were performed following the same proceeding described above for the EBOV trVLPs.

## Nanoluciferase experiments

EBOV trVLPs were produced containing a minigenome that encoded Nanoluciferase rather than Renilla luciferase using the protocol previously described for EBOV trVLP production under section heading Ebola transcription- and replication-competent virus-like particle (trVLP) assays. Target cells were plated in a 96 well format and spinfected with 200μl of cleared p0 supernatant the next day. Cells were then washed three times in medium 6 hours after infection. Cells were lysed with passive lysis buffer 48 hours after infection and reporter activity was then measured with a Nano-Glo luciferase assay system (Promega). For IFN-I based experiments increasing amounts of IFN-I were added 16 hours before infection and for TRIM25 overexpression experiments cells were reverse transfected with either TRIM25 or YFP as well as TIM1 24 hours before infection.

## Reporter gene assays

For transient reporter gene assays, HEK293T cells were seeded on 24-well plates and transfected with 100ng of control pcDNA-DEST40-GFP or pcDNA-DEST40 expressing individual ISG of interest in combination with 10ng of 3xkB-pCONA-Fluc [95], or pTK-ISG56-Luc (kindly provided by Greg Towers, UCL), or pNL(NlucP/ISRE/Hygro) (Promega) luciferase reporters and 10ng of pCMV-RLuc [95]. Cells were lysed with 1x Passive Lysis buffer 24 or 48 hours post-transfection, and Firefly and Renilla luciferase activities in the lysates were measured using a dual luciferase kit (Promega).

## Influenza A Minigenome assay

Influenza A polymerase activity was determined using a minigenome reporter containing the Firefly luciferase gene in a negative sense, flanked by the non-coding regions of the influenza NS gene segment transcribed from a species-specific pol I plasmid with a mouse terminator sequence [96,97]. Each viral polymerase component was expressed from separate pCAGGS plasmids encoding A/H1N1/Eng/195 NP, PA, PB1 and PB2 [98]. To analyse polymerase activity, HEK293T LacZ, HEK293T RIG-I CRISPR and HEK293T ZAP CRISPR cells were seeded in 24-well plates 24 hours prior transfection with 10ng PB1, 10ng PB2, 5ng PA, 15ng NP, 10ng pHPMO1-Firefly, 2.5ng pCAGGS-Renilla (transfection control, [99]) and increasing amounts of pcDNA3.1-TRIM25 using Trans-IT LT1. Cells were lysed 24 hours post-transfection with 1x Passive Lysis and reporter activities measured using a dual luciferase kit (Promega).

## Lentiviral assay

HIV-based lentiviral VLPs expressing GFP were produced on HEK293T cells seeded on a 10cm dish, and transfected with 6μg of pHR' Sin CSGW [100], 4 μg of pCRV-HIV Gag Pol and 2μg of pCMV-VSV-G using TransIT-LT1. Medium was changed 4–6 hours post transfection and VLP-containing supernatants harvested 48 hours later and cleared by centrifugation (5 minutes, 1200rpm).

U87-MG LacZ CRISPR control and U87-MG TRIM25 CRISPR KO cells were treated overnight with increasing amounts of IFN-I and then transduced with a fixed amount of pCSGW-HIV GP-VSVG VLPs prepared as above. 48 hours post-transduction cells were harvested and analysed for GFP expression by flow cytometry.

## Blam assay

EBOV BlamVP40-EBOV GP and EBOVBlaVP40-VSVg VLPs were produced on HEK293T cells. For that purpose, cells were plated in 10cm dishes and transfected with 6μg of pcDNA3.1-EBOV BlaVP40 [101] and 15μg of either pCAGGS-EBOV GP [102] or pCMV-VSVG using PEI max, and media was changed to OptiMEM 4–6 hours post-transfection. Supernatants were harvested past 48 hours and cleared by centrifugation (5 minutes, 1500rpm) before storage.

HEK293TIM-1 cells pre-treated with IFN-I, or transfected with either GFP or TRIM25, were 24 hours later transduced with Blam VLPs prepared as above, and incubated for 3–5 hours. After incubation cells were washed once with RPMI 1640 (no phenol red) medium (RPMI), and incubated in 1x loading solution of LiveBLAzer FRET-B/G Loading Kit (Life Technologies) with CCF2-AM for 1 hour (dark, room temperature). Following, cells were washed with 2.5mM Probenecid (Sigma) in RPMI and incubated in the same medium overnight (dark, room temperature). Next day, cells were analyzed by flow cytometry using BD FACSCanto II. Cells were gated on live cells and analyzed for CCF2 cleavage as readout for Blam VLP entry with Pacific Blue (cleaved) and FITC (not cleaved) laser channels, and using FlowJo 10.4.2 software for analysis.

## Production and purification of anti-EBOV antibodies

Rabbit antibodies against Ebola virus proteins NP, VP35 and VP30 were produced by Lampire Biological Laboratories using their standard protocol. In brief, two rabbits per each viral peptide were immunized with 0.5mg of antigen mixed with 0.5ml of Complete Freund's adjuvant. Two further boosts were performed in the same conditions at days 21 and 42. Serums used in this manuscript were collected by day 50. The sequences of the viral protein peptides used in the immunizations are as follow:

NP: DEDDEDTKPVPNRSTKGGQ
VP35: EAYWAEHGQPPPGPSLYEE
VP30: QLNITAPKDSRLANPTADD

For microscopy staining of the Ebola NP protein (see below), we have purified a fraction of the serum collected from rabbits immunized with the viral NP peptide. For that purpose, protein A beads (Invitrogen) were equilibrated by washing them with PBS, and then incubated with the serum for 2 hours at room temperature. Following, the beads were added into a column cartridge and washed 3 times with 10ml of PBS. IgGs were then eluted with 0.1M Glycine pH 3.5 in 5 batches of 1ml. IgG eluates were adjusted to pH 7.0 with 2M Tris Base, pH 9.0, and concentrated on Amicon Ultra centrifugal filter units (MWCO 10kDa) (SIGMA) with buffer exchanged to PBS. Approximate concentration of purified antibodies was determined by spectrometry (280nm) and fragments analyzed on SDS-PAGE.

### Analysis of cellular and viral proteins expression

Cells used in this study were seeded on 24-well plates, and transfected the following day with EBOV RNP-expressing plasmids as described above, either in combination with 150ng of individual ISG-expressing plasmids per well or followed by the addition 4 hours post-transfection of increasing amounts of type-I interferon. 24 hours later cellular lysates were subjected to SDS-PAGE and western blots performed using mouse monoclonal antibodies anti-HSP90 (Santa Cruz), anti-HA (Covance) or anti-TRIM25 (Abcam), or rabbit antibodies anti-EBOV NP, anti-EBOV VP35, anti-EBOV VP30, anti-TRIM25 (Abcam), anti-RIG-I (Enzo), anti-MAVS/VISA (Bethyl), anti-TBK1 (Abcam), anti-ZCCHV/ZAP (Abcam) or anti-NPC1 (Thermo Fisher). Visualizations were done by Image-Quant using either HRP-linked anti-mouse or anti-rabbit secondary antibodies (Cell Signaling).

### Immunoprecipitations

To address the interaction of TRIM25 with EBOV-NP proteins cells were seeded in 6-well plates and transfected with 500ng of either pcDNA3.1-GFP or pcDNA3.1-TRIM25wt, or mutants thereof, in combination with 125ng of pCAGGS-NP and/or 125ng of pCAGGS-VP35. At 24 hours post-transfection cells were lysed with a RIPA buffer containing 50mM Tris-HCl (pH 7.4), 150mM NaCl, 0.1% SDS, 0.5% sodium deoxycholate, 1% NP-40 and protease inhibitors (Roche). For immunoprecipitations performed with a VP35 antibody cells were lysed in 50mM Tris-HCL pH7.4, 1% NP40, 150mM NaCl, 1mM EDTA, 0.5% sodium deoxycholate and protease inhibitors. Following sonication, lysates were immunoprecipitated overnight with either rabbit anti-TRIM25 (Abcam) or rabbit anti-EBOV NP antibodies and protein G beads at 4°C on a tube rotator. On specific experiments, cellular lysates were treated prior immunoprecipitation at 37°C with 10μg/ml of RNase A (SIGMA) for 2 hours, or 1U/ml of RNAse III (Invitrogen) for 1 hour.

For ubiquitination-related pull-downs performed in the presence of exogenously expressed HA-ubiquitin cells were plated similarly in 6-well plates and transfected with 500ng of either pcDNA3.1-GFP or pcDNA3.1-TRIM25wt, in combination with 125ng of pCAGGS-VP35, 90ng of pCAGGS-VP30 and 1.2μg pCAGGS-L together with 125ng of pCAGGS-NP and/or 100ng of pMT123-HA-Ub [103]. At 24 hours post-transfection cells were lysed with a buffer containing 50mM Tris-HCl (pH 7.4), 150mM NaCl, 5mM EDTA, 5% Glycerol, 1% Triton-X 100, 10mM N-Ethylmaleimide (NEM) and protease inhibitors. Lysates were treated as above and immunoprecipitated overnight either with rabbit anti-EBOV-NP or mouse anti-HA (Covance). For ubiquitination-related pull-downs performed under endogenous levels of Ubiquitin, cells were plated in 6-well plates and transfected as above except for pMT123-HA-Ub. Cells were treated for 8 hours with 100nM of Bafilomycin A prior to lysis with a buffer containing 50mM Tris-HCl (pH 7.4), 150mM NaCl, 5mM EDTA, 5% Glycerol, 1% Igepal CA-630, 0.5% Sodium deoxycholate, 0.1% SDS, 50mM N-Ethylmaleimide (NEM) and protease inhibitors. Lysates were immunoprecipitated with an anti-EBOV NP antibody.

Cell lysates and pull-down samples were subjected to SDS-PAGE and Western blots performed using antibodies against HSP90 (Santa Cruz), TRIM25 (Abcam), EBOV-NP, EBOV-VP35, HA-tag (Rockland), ZAP (Abcam) and Ubiquitin. Blots were visualized by ImageQuant using anti-mouse or anti-rabbit HRP-linked antibodies (Cell Signaling).

### RNA immunoprecipitation

Cells were plated on 10cm dishes and transfected in the absence of EBOV RNP expressing plasmids with 2.5μg of pcDNA3.1-GFP, pcDNA3.1-TRIM25wt (or mutants thereof), or pcDNA4-ZAP-L [65] and medium changed 4–6 hours later. Alternatively, cells were treated

overnight with 1000U/ml of IFN-I or 250μM of T705. The following day cells were infected for 3 hours with 3ml of wild-type- or CpG low-EBOV trVLPs, or alternatively with the same volume of EBOV monocistronic VLPs, and then gently washed in PBS prior to 'on dish' irradiation with 400mJ/cm2 using a UV Stralinker 2400. Cells were subsequently lysed in 1ml of RIPA buffer (see above for composition) and sonicated. Cleared lysates were immunoprecipitated overnight at 4°C with rabbit anti-EBOV NP or rabbit anti-ZAP antibodies and protein G beads. Following 3 washes with RIPA buffer, the beads were resuspended in 200μl of RIPA and boiled for 10min to decouple protein/RNA complexes from the beads. Finally, input and pull-down samples were incubated with proteinase K (Thermo Fisher, 2mg/ml) for 1hr at 37°C, and then boiled for 10 minutes to inactivate the enzyme. Samples were stored at -20°C for downstream processing.

## RNA purification and quantitative RT-PCR

Total RNA was isolated and purified from EBOV trVLP infected cells using a QIAGEN RNAeasy kit, while viral RNA was extracted from supernatants using a QIAGEN QIAmp Viral mini kit, both accordingly to the manufacturer's instructions. Unless otherwise stated, 50ng of purified RNA was reverse transcribed by random hexamer primers using a High-Capacity cDNA Reverse Transcription kit (Applied Biosystems). Alternatively, strand-specific reverse transcription was performed as shown before [28] to generate cDNAs for viral genomic RNA using a reverse primer directed against the trailer region of the genome (-vRNA, EBOV -vRNA RT primer), for complementary RNA using a forward primer targeting the trailer region as well (+cRNA, EBOV +cRNA RT primer), or viral mRNA using oligo dT. Of the reaction, 5μl were subjected to quantitative PCR using primer/probe sets for human *Gapdh* (Applied Biosystems), and EBOV trVLP 5'-trailer region, L-Pol RNA or VP40 RNA (primer/probe sequences in S4 Table). Quantitative PCRs were performed on a QuantStudio 5 System (Thermo Fisher) and absolute quantification data analyzed using Thermo Fisher's Cloud Connect online software.

Input and pulldown samples from RNA immunoprecipitations were first resuspended in QIAzol (QIAGEN), and passed through QIAshredder columns (QIAGEN) for homogenization, and then passed to phase lock gel tubes (VWR) prior to addition of chloroform (SIGMA). After manually shaking the tubes, samples were centrifuged full-speed, for 15min at 4°C. The aqueous phase was passed to a new tube, and isopropanol added. After 10 minutes at room temperature, tubes were centrifuged as before and supernatants removed. RNA pellets were subsequently washed with 75% Ethanol, and spun at 7500xg, for 5 minutes at 4°C. Following aspiration of the supernatants, RNA pellets were left to dry and then resuspended in RNase-free water. Downstream processing for reverse transcription and quantitative RT-PCR was done as detailed above.

## Deubiquitinase assays

Prior to the deubiquitinase reactions, immunoprecipitations were performed using similar conditions to the ones described above for ubiquitination-related pull-downs. Briefly, HEK293T TRIM25 CRISPR KO cells were plated in 6 well plates and transfected with 500ng of either pcDNA3.1-YFP or pcDNA3.1-TRIM25$^{CR}$ in combination with 125ng of pCAGGS-VP35, 90ng of pCAGGS-VP30, 1.2μg of pCAGGS-L and 125ng of pCAGGS-NP, in the absence of ectopically expressed ubiquitin. Cells were lysed 48 hours post-transfection and lysates immunoprecipitated overnight with rabbit anti-EBOV-NP antibody using protein G agarose beads. Following immunoprecipitation, beads were treated with UbiCREST Deubiquitinase Enzyme Kit (BostonBiochem) following manufacturer's instructions. Briefly, beads

were washed several times before being equally split into separate tubes and treated with either USP2 or left untreated (NTC, non-treated control), for 30 minutes at 37˚C. Input lysates and pull-down beads were subjected to SDS-PAGE and Western blots performed using antibodies against HSP90 (Santa Cruz), TRIM25 (Abcam) and EBOV-NP. Blots were visualized by ImageQuant using anti-mouse or anti-rabbit HRP-linked antibodies (Cell Signaling).

### Microscopy

Cells were seeded on 24-well plates on top of coverslips pre-treated with poly-L-lysine to improve their adherence. For transient assays, cells were transfected with different combinations of EBOV RNP plasmids using the amounts described previously. Alternatively, cells were infected with 30µl of concentrated EBOV trVLPs in the absence of ectopically expressed EBOV RNP components.

Cells were fixed 24 hours post-transfection, or 4–6 hours post-infection, with 4% paraformaldehyde for 15 minutes at room temperature, and then washed first with PBS, followed by a second wash with 10mM Glycine. Next, cells were permeabilized for 15 minutes with a PBS solution complemented with 1% BSA and 0.1% Triton-X. Subsequently, cells were stained with mouse anti-TRIM25 (Abcam,10µg/ml) and purified rabbit anti-EBOV NP (1:50) antibodies diluted in PBS/0.01% Triton-X for an hour at room temperature. Cells were washed 3 times in PBS/0.01% Triton-X, followed by an incubation with Alexa Fluor 594 anti-mouse and Alexa Fluor 488 anti-rabbit antibodies (Molecular Probes, 1:500 in PBS/0.01% Triton-X) for 45 minutes in the dark. Finally, coverslips were washed once again 3 times with PBS/0,01% Triton-X and then mounted on slides with Prolong Diamond Antifade Mountant with DAPI (Invitrogen). Imaging was performed on a Nikon Eclipse Ti Inverted Microscope, equipped with a Yokogawa CSU/X1-spinning disk unit, under 60-100x objectives and Laser wavelengths of 405nm, 488nm and 561nm. Image processing and co-localization analysis was performed with NIS Elements Viewer and Image J (Fiji) software.

### Statistical analysis

Statistical significance was determined using paired two-tailed t tests calculated using the Prism software. Significance was ascribed to p values as follows: $^*p > 0.05$, $^{**}p > 0.01$, and $^{***}p > 0.001$. Data relative to viral based assays, signalling reporter assay and Blam assays were performed in duplicate in at least 3 independent experiments, and error bars represent +/-SEM.

For RNA IP experiments each data point is represented as average of three independent experiments +/-SD.

### Contact for reagents and resources sharing

Further information and requests for resources and reagents should be directed to and will be fulfilled by the Lead Contact, Stuart J.D. Neil (stuart.neil@kcl.ac.uk). Distribution of the CRISPR cell lines, antibodies, EBOV p4cis-CpG low-vRNA-Rluc and pT7-1cis-vRNA CpG low-EBOV-hrluc minigenomes generated in the course of this work will require signing of Material Transfer Agreements (MTA) in accordance with policies of King's College London. All requests for other trVLP components should be directed to Thomas Hoenen (thomas.hoenen@fli.de).

### Supporting information

**S1 Fig. (related to Fig 1).** EBOV trVLP assay and characterization of top candidate inhibitory ISGs. (**A**) U87-MG-TIM1 cells were transfected with plasmids expressing EBOV's RNP

components and pre-treated with increasing amounts of Universal type-I IFN-α, IFN-α2a or IFN-ß1a, 16–24 hours prior to EBOV trVLP infection. Reporter activities were measured 24 hours later. **(B)** U87-MG-TIM1 and HEK293T-TIM1 cells were transfected with plasmids expressing EBOV NP, VP35, VP30 and a HA-tagged version of L-polymerase, before being treated with increasing amounts of IFN-I prior to EBOV trVLP infection as in Fig 1B. Cells were lysed 24 hours post-transfection and lysates analysed by western blot for the expression of the EBOV RNP proteins and HSP90. **(C)** ISG Screen Validation. Normal distribution of EBOV trVLP Rluc values in the presence of individually over-expressed ISGs on p1 (left panel, blue dots) and p2 (right panel, red dots) target cells, normalized to Fluc levels on the same well. The number (n) of ISGs within standard deviation (s.d. or z score) ranges is shown in the boxes. **(D)** HEK293T-TIM1 cells were transfected with plasmids expressing the EBOV RNP proteins together with selected ISGs. Cells lysates were analysed 48 hours later by western blot as in (B). **(E)** HEK293T-TIM1 cells were transfected with individual ISGs and EBOV RNP-expressing plasmids as in (D), and tested for cellular viability 48 hours later using Cell-Titer Glo luminescence-based assay (Promega). As positive control for toxicity (red bar), cells were treated for 48h with a concentration of MG132 (25μM) sufficient to reduce the ATP levels in the supernatant by 50 percent. Values are represented as percentage of luminescence obtained in control wells transfected with GFP (grey bar), and toxicity cut-off represented as a dashed line. **(F)** Fold activation of Firefly luciferase NF-kB, ISG56/IFIT1 or ISRE reporters (top, middle or bottom panels, respectively) in HEK293T cells transiently transfected with selected individual ISGs compared to control GFP vector. Cells were harvested either 24 hours (blue) or 48 hours (grey) post-transfection.
(TIF)

**S2 Fig. (related to Fig 4).** Overexpression of ZAP-L impacts on EBOV trVLP viral RNA levels. **(A)** Quantification of viral RNA transcripts present intracellularly (left panel) and in supernatants (right panel) of HEK293T-TIM1 cells transfected as in Fig 4A, and infected with a fixed amount of EBOV trVLPs. Strand-specific reverse transcription primers were used on total RNA extracted from cells to generate cDNAs for minigenomic RNA (vRNA), complementary RNA (cRNA), and mRNA, followed by RT-qPCR analysis. Random hexamer primers were used to generate cDNAs from total RNA extracted from supernatants, and qPCR analysis performed using primers targeting the 5' trailer region of the 4cis minigenome or VP40 RNA. **(B)** EBOV trVLP reporter activities on U87-MG LacZ CRISPR-TIM1 (solid lines) and U87-MG ZAP CRISPR KO-TIM1 (dashed lines) target cells transfected with EBOV RNP proteins and pre-treated with increasing amounts of IFN-α2a (blue) or IFN-ß1b (red) prior to infection. **(C)** HEK293T LacZ CRISPR (grey) and HEK293T ZAP CRISPR KO (red) were used as producer cells (p0) for EBOV trVLPs, and reporter activities measured 48 hours post-transfection. Supernatants from p0 were used to infect HEK293T-TIM1 target cells (p1), and reporter activities determined 24 hours later. **(D)** Proteins levels of HSP90, ZAP and TRIM25 in lysates from Fig 4E were analysed by western blot on p1 cells harvest at time of infection. **(E)** Influenza A minigenome assay was performed in HEK293T LacZ CRISPR (grey), HEK293T ZAP CRISPR KO (red) and HEK293T RIG-I CRISPR KO (blue) cells transfected with Influenza polymerase components (NP, PB1, PB2 and PA) and increasing amounts of TRIM25. Normalized Fluc values are presented as percentage relative to a GFP control in each cell line. **(F)** HEK293T LacZ CRISPR, and HEK293T TRIM25 CRISPR KO cells were transfected with plasmids expressing the EBOV NP, VP35, VP30 proteins and a HA-tagged L Polymerase together with either pcDNA4 or ZAP-L. Cells lysates were analysed 48 hours later by western blot for the expression of HSP90, TRIM25, ZAP and the EBOV RNP complex proteins. **(G)** Relative quantification of intracellular viral RNA transcripts on cell lysates of HEK293T LacZ

CRISPR-TIM1 and HEK293T TRIM25 CRISPR KO-TIM1 cells transfected EBOV RNP plasmids in combination with either with GFP (grey) or ZAP-L (red), and infected with a fixed amount of EBOV trVLPs. Random hexamer primers were used to generate cDNAs from total RNA, and RT-qPCR analysis performed using a primers/probe sets targeting VP40 (mRNA, cRNA and vRNA, left), Trailer (cRNA and vRNA, right) and *gapdh*. Data normalized to HEK293T LacZ CRISPR-TIM1 cells transfected with GFP based on ΔΔCt values. **(H)** Graphical representation of the ratio between observed and expected CpG dinucleotide frequencies in the full-length EBOV genomic RNA (grey), in the wild-type trVLP 4cis genome (dark blue) and in the trVLP genomic variant with no CpG dinucleotides in the <u>Renilla</u> reporter sequence (CpG low, light blue). **(I)** Graphical representation of CpG dinucleotides localization in full-length EBOV genome (upper panel), in EBOV trVLP 4cis genome (middle panel) and in a 4cis genome with no CpG on the *Renilla* ORF (lower panel). CpG dinucleotides present on intergenic regions are represented in blue, while the ones present in viral ORFs are represented in red. CpG dinucleotides present on the *Renilla* reporter gene are represented in yellow. **(J)** U87-MG LacZ CRISPR and U87-MG KHNYN CRISPR KO cells were transfected with RNP proteins and TIM1, followed by an IFN-I pre-treatment prior to infection with a fixed amount of EBOV trVLPs. EBOV trVLP reporter activities in p1 were measured 24 hours after infection. **(K)** Relative quantification of intracellular viral RNA levels in cellular lysates from (I). Random hexamer primers were used to generate cDNAs and RT-qPCR analysis was performed using qPCR primers/probe sets targeting the VP40 RNA (vRNA, cRNA and mRNA, left panel) or the 5'-trailer region of the trVLP 4cis minigenome (vRNA and cRNA, right panel). Data are presented as fold change compared to control (no IFN) based on absolute copy numbers.
(TIF)

**S3 Fig. (related to Fig 5).** TRIM25-mediated ubiquitination of EBOV NP is independent of ZAP and MAVS. **(A)** Lysates of HEK293T-TIM1 cells transfected either with GFP or TRIM25, in combination with EBOV NP and/or EBOV VP35 were immunoprecipitated with an anti-NP antibody. Cellular lysates and pull-downs were analysed by western blot for HSP90, TRIM25, EBOV NP and VP35. **(B)** Lysates of HEK293T-TIM1 cells transfected either with EBOV NP and/or TRIM25 were left untreated (NT), or treated with RNase A (left panel) or RNAse III (right panel) prior to immunoprecipitation with an anti-TRIM25 antibody. Cellular lysates and pull-downs were analysed by western blot for HSP90, TRIM25 and EBOV NP. **(C)** Lysates of HEK293T LacZ CRISPR and HEK293T ZAP CRISPR KO cells transfected either with GFP or TRIM25 and/or EBOV NP were immunoprecipitated with anti-TRIM25 antibody. Cellular lysates and pull-down samples were analysed by western blot for HSP90, TRIM25, EBOV NP and ZAP. **(D)** HEK293T-TIM1 cells were transfected either with GFP, TRIM25 wild-type or TRIM25 ΔRING mutant, in combination with EBOV NP and/or a plasmid expressing a HA-tagged Ubiquitin (HA-Ub). Lysates from these cells were immunoprecipitated 48 hours later with an anti-HA antibody. Cellular lysates and pull-down samples were analysed by western blot for HSP90, TRIM25, NP and HA (ubiquitin). **(E)** HEK293T LacZ CRISPR and HEK293T ZAP CRISPR KO cells were transfected either with GFP or TRIM25, in combination with EBOV NP and/or HA-Ub. Lysates from these cells were immunoprecipitated 48 hours later with an anti-HA antibody. Cellular lysates and pull-down samples were analysed by western blot as in (D). **(F)** HEK293T LacZ control and HEK293T FL-MAVS/miniMAVS CRISPR DKO cells were transfected either with GFP or TRIM25 in combination with EBOV NP and/or HA-Ub. Lysates were immunoprecipitated with anti-HA antibody and analysed by western blotting as in (D). **(G)** HEK293T TRIM25 CRISPR KO cells were transfected either with GFP or TRIM25 CR in combination with EBOV NP and/or

HA-Ub. Lysates were immunoprecipitated with anti-HA antibody and analysed by western blotting as in (D). **(H)** HEK293T LacZ CRISPR control and HEK293T TRIM25 CRISPR KO cells were infected with a fixed volume of EBOV trVLPs concentrated on a 20% sucrose-cushion. Cell were lysed at the depicted time points after infection and lysates analysed by western blot for HSP90 and EBOV NP.
(TIF)

**S4 Fig. (related to Fig 7).** EBOV trVLP replication is dependent on the entry factor NPC1 and is sensitive to transcription inhibitor T705 (Favipiravir). **(A)** Cellular lysates from HEK293T LacZ CRISPR-TIM1, HEK293T NPC1 CRISPR KO-TIM1 and HEK293T NPC1 CRISPR KO-TIM1 cells with restored expression of NPC1 (+NPC1) were analysed by western blot for HSP90 and NPC1. **(B)** The cells lines used in (A) were transduced either with BlaVP40-EBOV-GP or BlaVP40-VSV-G virus-like particles, and 24 hours later the percentages of cells presenting cleavage of CCF2-AM dye were determined by flow cytometry as readout for viral particle entry. **(C)** EBOV trVLP normalized reporter activity on the upper mentioned cell lines (p1) transfected with EBOV RNP proteins and infected the following day with a fixed amount of EBOV trVLPs. Reporter activities were measured 24 hours later. **(D)** Quantification of intracellular viral RNA levels in HEK293T-TIM1 target cells 3 and 24 hours post-infection with a fixed amount of EBOV trVLPs. Prior to infection the cells were transfected with EBOV RNP expressing plasmids together with GFP (grey), TRIM25 (blue) or ZAP-L (red). Strand-specific reverse transcription primers were used on total RNA extracted from cells to generate cDNAs for minigenomic RNA (vRNA), complementary RNA (cRNA), and mRNA, followed by RT-qPCR analysis. **(E)** HEK293T LacZ CRISPR-TIM1 cells were transfected with GFP, TRIM25 or ZAP-L prior to infection with EBOV trVLPs. 3 hours post-infection cells were UV cross-linked, and EBOV NP from incoming virions were immunoprecipitated from lysates with an anti-NP antibody. Cellular lysates and pull-down samples were analysed by western blot for HSP90, TRIM25, NP and ZAP. **(F)** Relative quantification of NP-associated RNA in HEK293T LacZ CRISPR-TIM1 cells transfected with GFP (grey), TRIM25 (blue) or ZAP-L (red) prior to infection with wild-type or CpG low EBOV trVLPs. 3 hours post-infection cells were UV cross-linked, and EBOV NP from incoming virions was immunoprecipitated from lysates with an anti-NP antibody. Following proteinase K treatment, pulled-down RNAs were extracted with Qiazol / chloroform, random hexamer primers were used to generate cDNAs, and RT-qPCR analysis performed using a primers/probe set targeting EBOV VP40 RNA. Values are presented as a percentage of absolute RNA copy numbers on cells transfected with GFP. **(F)** HEK293T LacZ CRISPR, HEK293T ZAP CRISPR KO and HEK293T TRIM25 CRISPR KO cells stably expressing TIM1 were transfected with GFP (grey), TRIM25 (blue) or ZAP-L (red) as depicted in the panels, and later infected with wild-type (solid bars) or CpG low (striped bars) EBOV trVLPs. Relative quantification of NP-associated RNA was determined as in (E). **(H)** HEK293T LacZ CRISPR-TIM1 cells were transfected with GFP, TRIM25, ΔRING or ΔSPRY prior to infection with EBOV trVLPs. 3 hours post-infection cells were UV cross-linked, and EBOV NP from incoming virions were immunoprecipitated from lysates with an anti-NP antibody. Cellular lysates and pull-down samples were analysed by western blot for HSP90, TRIM25 and NP. **(I)** HEK293T TRIM25 CRISPR KO-TIM1 cells were transfected with GFP or CRISPR-resistant versions of TRIM25wt, or mutants thereof. 3 hours post-infection cells were UV cross-linked, and EBOV NP from incoming virions were immunoprecipitated from lysates with an anti-NP antibody. Cellular lysates and pull-down samples were analysed by western blot for HSP90, TRIM25 and NP. **(J)** EBOV trVLP reporter activities on HEK293T-TIM1 target cells (p1), transfected with RNP proteins and pre-treated overnight with increasing amounts of T-705 (Favipiravir) prior to infection. Reporter activities were

measured 24 hours after infection. **(K)** Relative quantification of intracellular viral RNA levels in HEK293T-TIM1 p1 target cells pre-treated with increasing amounts of T-705 compound and infected with EBOV trVLPs as in (J). Random hexamer primers were used to generate cDNAs and RT-qPCR analysis was performed using qPCR primers/probe sets targeting the VP40 mRNA or *gapdh* as endogenous control. Data presented as fold change compared to control (no T-705) based on ΔΔCt values. **(L)** Graphical representation of CpG dinucleotides localization on a monocistronic genome containing a *Renilla* reporter gene flanked by the 5'-leader and 3'-trailer regions of EBOV genome (upper panel), and its Low-CpG variant (bottom panel) in which all CpGs in the *Renilla* ORF were silently mutated. CpG dinucleotides present on the trailer and leader regions are represented in blue, while in yellow are the CpG dinucleotides present in the *Renilla* reporter gene. **(M)** Normalized reporter activity of the monocistronic genomes on HEK293T-TIM1 target transfected with EBOV RNP proteins, and either GFP (grey) or ZAP-L (red) prior to infection with increasing amounts of wild-type monocistronic VLPs (WT, solid lines) or a variant with no CpG dinucleotides on the *Renilla* ORF (CpG low, striped lines). **(N)** HEK293T LacZ CRISPR-TIM1 cells were transfected with GFP (grey) or ZAP-L (red) and later infected with wild-type (solid bars) or CpG low (striped bars) monocistronic VLPs. Relative quantification of ZAP-associated RNA was determined as in (F). **(O)** HEK293T LacZ CRISPR and HEK293T NPC1 CRISPR stably expressing TIM1 were infected with either WT or ΔGP EBOV nanoluciferase trVLPs. EBOV trVLP nanoluc reporter activities were measured 48 hours post-infection. **(P)** HEK293T TRIM25 CRISPR KO cells were transfected with TIM1 and either a CRISPR-resistant version of TRIM25 or YFP before infection with WT or ΔGP EBOV nanoluciferase trVLPs. EBOV trVLP nanoluc reporter activities were measured 48 hours post-infection.
(TIF)

**S1 Table. Large Scale ISG Screen: EboV trVLP in HEK293T cells–p1 and p2 values.**
(PDF)

**S2 Table. List of CRISPR Guides used in the study.**
(DOCX)

**S3 Table. List of primers used for cloning in the study.**
(DOCX)

**S4 Table. Primers and probes used for cDNA synthesis and RT-qPCR.**
(DOCX)

## Acknowledgments

We thank other members of the Neil and Swanson laboratories for helpful discussions and Wendy S Barclay (Imperial College London) for materials. We are grateful to Stefan Becker and his group, as well as Adam Fletcher for advice.

## Author Contributions

**Conceptualization:** Rui Pedro Galão, Harry Wilson, Thomas Hoenen, Sam J. Wilson, Chad M. Swanson, Stuart J. D. Neil.

**Data curation:** Rui Pedro Galão, Harry Wilson, Kristina L. Schierhorn.

**Formal analysis:** Rui Pedro Galão, Harry Wilson, Kristina L. Schierhorn, Franka Debeljak, Bianca S. Bodmer, Daniel Goldhill, Thomas Hoenen.

**Funding acquisition:** Thomas Hoenen, Sam J. Wilson, Stuart J. D. Neil.

**Investigation:** Rui Pedro Galão, Harry Wilson, Kristina L. Schierhorn, Franka Debeljak, Bianca S. Bodmer, Daniel Goldhill, Thomas Hoenen, Sam J. Wilson, Chad M. Swanson, Stuart J. D. Neil.

**Methodology:** Rui Pedro Galão, Harry Wilson, Kristina L. Schierhorn, Franka Debeljak, Bianca S. Bodmer, Daniel Goldhill, Thomas Hoenen, Sam J. Wilson, Chad M. Swanson, Stuart J. D. Neil.

**Project administration:** Rui Pedro Galão, Stuart J. D. Neil.

**Resources:** Thomas Hoenen, Stuart J. D. Neil.

**Supervision:** Rui Pedro Galão, Thomas Hoenen, Stuart J. D. Neil.

**Validation:** Rui Pedro Galão, Thomas Hoenen, Stuart J. D. Neil.

**Visualization:** Rui Pedro Galão, Harry Wilson.

**Writing – original draft:** Rui Pedro Galão, Harry Wilson, Stuart J. D. Neil.

**Writing – review & editing:** Rui Pedro Galão, Harry Wilson, Kristina L. Schierhorn, Franka Debeljak, Thomas Hoenen, Sam J. Wilson, Chad M. Swanson, Stuart J. D. Neil.

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
