## [Decision Letter · Decision Letter 0]

9 Aug 2021

Dear Dr Galao,

Thank you very much for submitting your manuscript "TRIM25 and ZAP target the Ebola virus ribonucleoprotein complex to mediate interferon-induced restriction" for consideration at PLOS Pathogens. As with all papers reviewed by the journal, your manuscript was reviewed by members of the editorial board and by several independent reviewers. In light of the reviews (below this email), we would like to invite the resubmission of a significantly-revised version that takes into account the reviewers' comments.  

We cannot make any decision about publication until we have seen the revised manuscript and your response to the reviewers' comments. Your revised manuscript is also likely to be sent to reviewers for further evaluation.

Sincerely,

Alexander Bukreyev, Ph.D.

Associate Editor

PLOS Pathogens

Christopher Basler

Section Editor

PLOS Pathogens

Kasturi Haldar

Editor-in-Chief

PLOS Pathogens

orcid.org/0000-0001-5065-158X

Michael Malim

Editor-in-Chief

PLOS Pathogens

orcid.org/0000-0002-7699-2064

Reviewer's Responses to Questions

**Part I - Summary**

Reviewer #1: The manuscript ‘TRIM25 and ZAP target the Ebola virus ribonucleoprotien complex to mediate interferon-induced restriction’ by Galao et al. examines the potential role of interferon-stimulated genes (ISGs) in restricting the replication and transcription of Ebola virus (EBOV) and describe TRIM25 and ZAP as ISGs that interdependently antagonize the replication and transcription of EBOV transcription and replication competent virus like particles (trVLP). The manuscript first describes the sensitivity of the trVLP system to type I interferon (IFN-I), and then identifies TRIM25 as host factor that antagonizes EBOV trVLP production. Subsequent experiments examined the mechanism through which TRIM25 acts, and the authors suggest that TRIM25 ubiquitinates EBOV nucleoprotein (NP), facilitating the dissociation of NP from the viral minigenome RNA allowing ZAP to bind to CpG rich regions of the minigenome.

In general, this manuscript presents a potentially important antiviral mechanism against Ebola virus, which would be of interest to the field. Although the paper presents evidence for interaction among TRIM25, ZAP, and EBOV vRNP in the context of an interferon-stimulated cell, the functional significance is lacking especially when considering the strong IFN-I inhibition during EBOV infection as well as the use of a surrogate VLP system. Although it is understandable the need of establishing model systems to study Ebola in lower biocontainment, this unfortunately diminishes the relevance of the study. There are also some inconsistencies, and controls for some experiments were not very carefully performed. Overall, the authors show a connection between ISGs ZAP and TRIM25 that could represent an important anti-EBOV mechanism but the function in the context of an IFN-I and EBOV replication is unclear.

Reviewer #2: In their manuscript entitled “TRIM25 and ZAP target the Ebola virus ribonucleoprotein complex to mediate interferon-induced restriction,” Galao and colleagues present evidence for a novel EBOV-specific restriction pathway mediated by the cellular proteins TRIM25 and ZAP. The authors use the EBOV transcription and replication competent virus like particle (trVLP) system to screen for IFN-induced restriction of trVLP production. From this screen the authors identify TRIM25 as a restriction factor, and they subsequently show that its activity is independent of RIG-I but dependent on MAVS. The authors further demonstrate that ZAP is necessary for the TRIM25-mediated restriction of EBOV trVLPS and that, together, these two proteins promote the ubiquitination of NP and its dissociation from the trVLP genome. The conclusions of this paper are interesting, and they may have important implications on the EBOV replication cycle. The authors should be commended for their thorough investigation and the significant amount of work depicted herein. Likewise, the manuscript is well organized and well written. Nevertheless, there are a few aspects of the manuscript that could be improved, as described below.

Reviewer #3: In this manuscript, the authors identified TRIM25 as a potent EBOV inhibitory host factor by use of EBOV trVLP-based interferon-stimulated genes screen. They demonstrated that TRIM25 and a related adaptor molecule ZAP contribute to type I IFN-mediated inhibition of replication process of trVLP. They also confirmed that TRIM25 interacts with the EBOV NP and promotes its ubiquitination, leading to dissociation of NP from the vRNAs. The authors further observed that ZAP targets the CpG-rich region of vRNA. Taken together, the authors concluded that TRIM25 facilitates interaction of vRNAs with ZAP by exposing CpG-rich vRNA region, resulting in suppression of trVLP replication. Although their findings suggest the potential contribution of TRIM25 in host defense against EBOV infection, there are several fundamental problems in the experimental design. Firstly, it is not clear why the authors selected a glioma cell line for evaluation of TRIM25 functions in trVLP system. Some critical experiments were not conducted appropriately. Moreover, the significance of TRIM25 and related adaptor molecules in the inhibition of trVLP replication was not robust. Please refer to my specific comments as follows:

**Part II – Major Issues: Key Experiments Required for Acceptance**

Reviewer #1: General points:

1) A main concern for this paper is that the roles of TRIM25 are only observed following pre-treatment with type I IFN. The authors claim while describing figure 2I that “in the absence of IFN treatment, TRIM25 knockout HEK293T cells gave significantly higher Rluc activity in p0 producer cells and yielded significant higher levels of virus (p1 infetivity)”, yet in Figures 2F and 2G there is no difference between WT and TRIM25 knockout cells in the absence of IFN-I. Also, the authors first identified TRIM25 through an over-expression screening that screened in the context of IFN-I treated cells. Since the phenotype is observed only in the presence of IFN, alternative possibility to the model described by the authors is that TRIM25 somehow affects the IFN signaling pathway (downstream of the IFN receptor), leading to effects on other ISGs that equally could affect EBOV replication. This is possibility is not ruled out. In addition, the authors should discuss in the manuscript when and how during infection the role for TRIM25 they are describing may be relevant. The levels of IFN-I for these studies to see a strong phenotype is also very high relative to physiological levels of IFN-I that are produced in response to a viral infection. Overall, the reliance on pre-stimulation with IFN-I or TRIM25 over-expression to see a phenotype bring in to question whether the findings reflect the phenotype one would observe during EBOV infection.

2) There is inaccurate description of EBOV VP35 as co-factor for NP. On at least two occasions (line 76 and 306) the authors state VP35 is NP’s co-factor. VP35 is the polymerase (L) co-factor (Tchesnokov et al. 2018, Muhlberger et al. 1999). VP35 does act as a chaperone for NP (Leung et al. 2015, Kirchdoerfer et al. 2015) and links NP and L to facilitate transcription and replication of the EBOV genome. Also, when the authors describe viral ribonucleoprotein they describe it as being composed of NP, VP30, VP35, and L, but this is the composition of the viral transcription complex. The viral RNP is composed of, minimally, NP, VP35, and VP24 (Huang et al. 2002).

3) In regard to the authors’ over-expression experiments, for several immunoprecipitations, the protein being pulled down is clearly not equal which may bias their interpretation of the results. For example, in Figure 5A, the authors are pulling down for endogenous TRIM25 and the levels of the input of TRIM25 are higher in the lanes expressing nucleoprotein than GFP or VP35 alone. Although the levels of TRIM25 in the PD are not that different, the levels of VP35 in the input are also lower than in the other sample. This is important since the authors are claiming that TRIM25 does not bind to VP35 in the absence of NP. The Figure S3A also has a similar issue where TRIM25 is not expressed as highly in the VP35 only sample as the NP only and NP + VP35 lanes (Input). Although the IP in the case of Figure S3A was for NP rather than TRIM25, it is odd that the authors would have obvious differences in expression in the corresponding samples in two different experiments. Similar issues of differences in expression of controls are also observed in Figure S3D, S3E, and S3F. The authors should do an immunoprecipitation for VP35 for NP, TRIM25, and NP + TRIM25 to be certain that there is interaction between VP35 and TRIM25

4) The authors also do not show evidence that TRIM25 directly ubiquitinates NP. They rely on over-expression of TRIM25 or HA-Ub, and they do not interrogate which lysine residues or region of NP is ubiquitinated or perform an in vitro ubiquitination assay to clearly demonstrate TRIM25 is directly ubiquitinating NP. No endogenous ubiquitination is shown either.

Specific Comments:

1) Lines 65-67: The paper referenced regarding evidence for bat hosts of EBOV is a serology-based studies. Stronger papers to support this statement in regard to EBOV include the finding of Bombali virus in insectivorous bats (ex. Goldstein et al. 2018).

2) Lines 76-89: In describing the vRNPs of EBOV the authors make a couple errors, as described above, in their reporting of VP35’s roles and including VP30 as an essential vRNP component while leaving out VP24.

3) The authors do not discuss the host’s IFN-I response to EBOV or filovirus infection in the introduction. Considering the phenotypes they observe in the context of the trVLP system are highly reliant of IFN-I pre-stimulation, it is important to provide this information.

4) It would be beneficial to include the Figure S1A in the main text and/or give a brief explanation of the trVLP system (specifically the minigenome component itself) in-text. The Figure S1A itself is also difficult to read.

5) Figure 1B: It is unclear why this data was presented on a log scale when the output is reported as a percentage. There is probably a more substantial effect of IFN-I on EBOV particle assembly that would be easier to see with a linear scale, especially considering the statistical test indicates a significant difference.

6) In the over-expression experiments for reporter assays (Figure 2A) and throughout the paper, the authors do not include western blots to monitor over-expression of TRIM25.

7) Lines 190-193 and Figure 2A: It is unclear why there would be an equivalent difference between p1 and p2. The authors should provide some explanation. How are the authors able to control how many trVLPs are added from the p1 supernatants to p2? If there is no way to control this, then I would expect there to be a larger difference in p2 than p1.

8) Figure 2I: As stated above, it is unclear why you observe a difference in trVLP production in p0 TRIM25 knockout cells when you do not see a difference between wt and TRIM25 knockout cells in the absence of IFN-I treatment in Figures 2E, F and 2G.

9) Lines 211-214: The authors should include a reference for this statement

10) Figure 4: Across the panels, it is not consistent whether ZAP-L has an effect in the absence of IFN-I. It’s unclear if this is due to the method of normalization or the lack of an effect in the absence of IFN-I. If it is due to the method of normalization, the authors explain more clearly.

11) It may be useful to include supplementary figures S4I and S4J in the main text.

12) Figures 5 and 6 were flipped in the document.

13) TRIM25 can bind both dsRNA and ssRNA (Sanchez etal. 2018), but RNAse A will only work for single-stranded RNA. Since there are dsRNA replication intermediates of the EBOV minigenome the experiment should be repeated with RNAse III which can cleave dsRNA.

14) Several western blots have background for NP (ex. Figure 5E) in the GFP pull-downs.

15) Figures 5E and S3D: The way the experiment is laid out, it is difficult to appreciate a difference in the ubiquitination of NP with the addition of TRIM25. It would be more effective to have the HA-Ub IP with TRIM25 WT + NP and TRIM25 catalytic mutant/RING domain deletion in adjacent lanes. In the supplementary figure there is less HA-Ub expressed and pulled down in the samples with the TRIM25 mutant as well further complicating an accurate interpretation.

16) Looking at endogenous levels of NP ubiquitination in wild-type versus TRIM25 knockout cells following infection with the trVLPs would be a clearer experiment. Especially if the knockout is rescued with wild-type or catalytic mutant TRIM25.

17) Figure 6A: It is unclear what groups are being compared for statistics.

18) Lines 342-345: The authors need to include citations regarding the TRIM25 RNA binding sites

19) Figure 6B: it appears that the authors assume that the NP smear corresponds to ubiquitinated NP (NP-Ubn). This needs to be shown with anti-ubiquitin antibody against endogenous Ub.

20) Figure 6C: The pull down for TRIM25 is not equal. Since the authors are claiming that the SPRY deletion mutant loses interaction, there needs to be an equivalent amount of TRIM25 expressed.

21) Lines 369-371: It would be helpful if the authors framed the 5% of cells with NP-TRIM25 interactions in the context of the percentage of cells that are NP positive.

22) Lines 439-445: Based on the data presented in the paper, I would argue that TRIM25 is recruited to viral RNA rather than NP since the RNA binding mutants also have decreased interaction with NP. The authors do state that RNA binding regions of TRIM25 could be influencing the conformation of TRIM25 SPRY and thus binding to NP, but further experiments are needed to clarify this aspect of their proposed model.

23) Lines 449-451: It is unclear why the authors mention that “until recently” TRIM25 is positive regulator of RIG-I yet later in their discussion point to TRIM25 being targeted by viral proteins (lines 560-562) in regard to their blocking TRIM25 from activating RIG-I.

24) Lines 549-552: The contribution of MAVS to TRIM25’s proposed antiviral function is not well explained.

25) Line 569: The authors should include a citation

26) Lines 856-868: The authors use digital droplet PCR and do not need a standard curve to determine DNA copies, however the authors need to include validation for strand specificity or provide reference to a paper that have previously done the validation with these primers.

Reviewer #2: 1. The majority of the data presented in this paper is the result of a few kinds of experiments repeated multiple times with varying conditions. For example, trVLP replication/transcription in p1 and p2 cells is assessed via luciferase reporter assay under conditions that include IFN treatment, ISG over-expression, gene knockout, etc. Similarly, trVLP replication/transcription is also assessed via RT-qPCR under similarly varying conditions. This means that much of the data in this paper is (or should be) consistent from experiment to experiment, outside of the effects of whatever variable is changed. However, it is confusing that none of the data is presented consistently. For the sake of clarity, it would be extremely helpful if the authors could revise their figures to standardize axes, titles, and assays throughout their paper. For example,

a. trVLP assays are depicted in Figures 1A, 2A, 2E, 2G, 2I, 3C-E, 3G, 3I, 3J, 4A, 4C, 4E, 4G, 4I, 5D, and 6A; however, none of these figures use the same y-axis range. Indeed, in the case of 4A, the y-axis range differs between the two panels in the same sub-figure. As a result, the magnitude of the differences between conditions within an experiment/figure cannot be easily compared with the results of a different experiment/figure. In some cases, such as in Figures 3C and 3G, the narrow y-axis range effectively increases the appearance of a difference in magnitude when the actual differences is quite small. Since all of these data are based on the same fundamental experiment, it seems appropriate to use identical y-axis ranges for all.

b. The figures depicting RT-qPCR data are also confusing at times. The methodology of Figures 1 C and D are ostensibly identical to that of Figure G, yet the axis labels and order of the data panels differs. Is it necessary to apply a different label to the vRNA supernatant data in Figures 1D and 1G? Is it necessary that these labels differ from the ones used in Figure 1C and the other two panels in Figure 1G? The presentation of data is also inconsistent in Figure 2F, 4D, and 4H. Please clarify.

c. Similarly, the y-axis ranges in Figures 7B-F and H should all be the same.

2. In Figure 2C, strand-specific RT-qPCR used to quantify the differences among vRNA, cRNA, and mRNA, but this was not done for any of the other RT-qPCR assays. Why?

3. Can the authors specify the dose of trVLPs used in each assay? On Line 155 they state that “a fixed dose” of trVLPs was used, but the dose is not stated. On Lines 675, they state that between 5-50 ul of supernatants were transferred to p1 cells. Why would the volume change from experiment to experiment? Do differences in the inoculum volume affect the differences observed in each of the assays? Please clarify.

4. One of the biggest weaknesses of this study is that the authors provide no evidence of TRIM25/ZAP-mediated restriction of authentic EBOV infection. Can the authors attempt to validate their results using authentic virus? Such an experiment would greatly enhance the relevance of the conclusions, in addition to demonstrating the “translatability” of the trVLP modelling system. It is understandable if this kind of experiment is not possible, since it would require access to a high-containment laboratory. However, the authors should at least acknowledge this shortcoming.

Reviewer #3: 1. Figures 2 and 3: Some experiments were performed with HEK293T or U87-MG cells alone, although the authors have established the CRISPR/Cas9-based KO clones for key factors in both cell types.

2. Figures 2, 3, 4, and 7: I suggest that the authors add the data showing the exogenously expressed genes.

3. Figures 5B and 7A: How do the authors set up the experiment condition? The information about the incubation time in the figure legend and the text is not consistent. The authors must provide the precise and detailed methodology in the manuscript. According to the pattern of NP signals in the images, they seem to be inclusion bodies instead of incoming single vRNPs.

**Part III – Minor Issues: Editorial and Data Presentation Modifications**

Reviewer #1: (No Response)

Reviewer #2: 1. Can the authors indicate (in the figure legends or Materials and Methods, for example) the amounts of IFN that were used in each assay instead of referring to “increasing amounts of IFN”?

2. Are the differences in RNA levels in Figure 4D significant?

3. On Line 370, the authors state that 5% of transfected cells exhibited NP foci that colocalized with TRIM25. Does this mean that 95% of the cells expressing NP did not exhibit the same colocalization? What might this say about the ability of TRIM25/ZAP to act as an EBOV restriction factor?

Reviewer #3: 1. Please describe the technical terms in the text and the figures in a consistent way (e.g. TIM-1 or TIM1, TRIM25 or Trim25, EBOV or EboV, hours post-infection or hpi, HA-Ub or Ub-HA).

2. Please indicate the origin of cell lines in the materials and methods. Please explain the reasons that the authors used HEK293T cells for ISG screen although U87-MG cells exhibited more significant sensitivity to IFN-I (Figure 1C and D).

3. Line 237, what is RLR signaling?

4. Line 298, Figure S4I and S4J must be S2I and S2J, respectively.

5. Figures 4G, 4I, S2D, and S2J: Please add the results of statistical analysis.

6. Figure1E-G: Please highlight TRIM25, IRF2 and BST2 in the figure.

7. Figure S1A, lower panel: The budding virions seem to be the vRNPs.

8. Figure 7C and H: Please modify the color of legends for NP-associated RNAs.

PLOS authors have the option to publish the peer review history of their article (what does this mean?). If published, this will include your full peer review and any attached files.

Reviewer #1: No

Reviewer #2: No

Reviewer #3: No
---

## [Decision Letter · Decision Letter 1]

3 Dec 2021

Dear Dr Galao,

Thank you very much for submitting your manuscript "TRIM25 and ZAP target the Ebola virus ribonucleoprotein complex to mediate interferon-induced restriction" for consideration at PLOS Pathogens. As with all papers reviewed by the journal, your manuscript was reviewed by members of the editorial board and by several independent reviewers. In light of the reviews (below this email), we would like to invite the resubmission of a significantly-revised version that takes into account the reviewers' comments.

Note that reviewers 1 and 3 still feel additional data is necessary. 

We cannot make any decision about publication until we have seen the revised manuscript and your response to the reviewers' comments. Your revised manuscript is also likely to be sent to reviewers for further evaluation.

Sincerely,

Alexander Bukreyev, Ph.D.

Associate Editor

PLOS Pathogens

Christopher Basler

Section Editor

PLOS Pathogens

Kasturi Haldar

Editor-in-Chief

PLOS Pathogens

orcid.org/0000-0001-5065-158X

Michael Malim

Editor-in-Chief

PLOS Pathogens

orcid.org/0000-0002-7699-2064

Reviewer's Responses to Questions

**Part I - Summary**

Reviewer #1: Overall, the Galao et al. resubmission of “TRIM25 and ZAP target the Ebola virus ribonucleoprotein complex to mediate interferon-induced restriction” is much improved from the initial submission, and the majority of comments were addressed sufficiently. However, two concerns remain to be addressed as described below. The main concern that must be address is definitive evidence that NP is ubiquitinated. One of the main conclusions is ubiquitination of NP and this has to be demonstrated by probing for endogenous ubiquitin. This is consistently done in the ubiquitin field, and there is no reasonable explanation on why technically it is not possible to use antibodies against endogenous Ub. Furthermore, TRIM25 is also responsible for ISGylation, and has also been reported to produce unanchored ubiquitin chains. The methods that the authors use cannot distinguish between these, thus they need to include an experiment using endogenous ubiquitin and an ubiquitin-specific antibody.

Reviewer #2: In the revised version of their manuscript, Galao and colleagues have adequately addressed almost all of the reviewer comments. The result is a much improved paper, particularly with respect to the presentation and description of the data. I have only a few minor comments that might further enhance clarity.

Reviewer #3: The authors have incorporated some comments from the first round of review; however the following key issues remains. Please refer to my specific comments in Part II – Major Issues.

**Part II – Major Issues: Key Experiments Required for Acceptance**

Reviewer #1: Since TRIM25 is both an ubiquitin and ISG15 ligating enzyme (3, 5, 6), it is essential that the authors provide definitive and direct evidence that NP is ubiquitinated versus ISG15ylated. Further, USP2 is a cross-reactive enzyme that is able to cleave both ubiquitin and ISG15 from modified proteins (7), thus the use of this particular deubiquitinase cannot be used to confirm that the modification to NP is ubiquitin. The authors must perform pull-down assays for NP (in denaturing conditions), and immunoblot for endogenous ubiquitin. Show a blot with ubiquitin-specific antibody, specifically in the absence of Ub over-expression. The author’s response to reviewer #1’s specific comments 16 and 19 are thus insufficient responses. Commercially available antibodies against endogenous ubiquitin are used extensively and are of excellent quality. There is really no reasonable explanation on why not to use antibodies against endogenous ubiquitin. This is standard in the field.

If the issue is NP, Commercial NP antibodies are available that may work better than the antibodies generated in-house (https://www.kerafast.com/item/1007/anti-zaire-ebola-virus-nucleoprotein-1e6-antibody). Figure 5E shows perfectly good pull-down for NP, so no reason why not to immunoblot for endogenous ubiquitin in the absence of over-expressed Ub. Please add molecular weigh markers to all blots.

Reviewer #2: None

Reviewer #3: 1. Although the authors advocated that they used U87-MG glioma cell line because of its sensitivity and resoinsiveness to the IFN signaling, glioma cells are not major target cells in the context of EBOV infection. In fact, quite few EBOV studies used this cell line in the literature. As the authors mentioned in the introduction (lines 111-112), the myeloid cell lines must be more appropriate to assess the roles of TRIM25 and ZAP in EBOV life cycle.

2. The authors argued that theoretically transcription of NP protein would not occur and all the detected NP signals in this experiment must be derived from incoming vRNPs. However, I still don’t understand the NP signal pattern in Figure 7. The size of each vRNP is supposed to be approximately 100 nm in diameter and 1-2 µm in length. Detected NP signals are quite large for its predicted size in these images. Why?

**Part III – Minor Issues: Editorial and Data Presentation Modifications**

Reviewer #1: 1) Please add Molecular weight markers to ALL immunoblots.

2) In response to the author’s response to reviewer #1’s main point 3, their explanation for why the experiments were not repeated to ensure controlled expression between groups for IPs is insufficient. It is known that TRIM25 undergoes autoubiquitination (1, 2) and autoisgylation (3), but in these studies and others TRIM25 expression (over-expression and endogenous) does not seems to be influenced regardless of infection, capacity for autoubiquitination, or over-expression of other proteins (2-4). Further, they provide no citations to support their claims and simply state ‘see literature’. The conclusions are unlikely to be changed substantially by the differences in expression, but the paper could be improved by having experiments with equal levels of expression.

3) In response to reviewer #1’s specific point 14, their explanation does not address the original comment adequately. The authors will need to show a GFP pull down the in TRIM25 knockout cells to demonstrate no NP pulldown background to support their claim in their response to the reviewer that it is due to endogenous TRIM25.

References

1. Gupta S, Ylä-Anttila P, Callegari S, Tsai MH, Delecluse HJ, Masucci MG. Herpesvirus deconjugases inhibit the IFN response by promoting TRIM25 autoubiquitination and functional inactivation of the RIG-I signalosome. PLoS pathogens. 2018;14(1).

2. Choudhury NR, Heikel G, Trubitsyna M, Kubik P, Nowak JS, Webb S, et al. RNA-binding activity of TRIM25 is mediated by its PRY/SPRY domain and is required for ubiquitination. BMC biology. 2017;15(1).

3. Zou W, Wang J, Zhang DE. Negative regulation of ISG15 E3 ligase EFP through its autoISGylation. Biochemical and biophysical research communications. 2007;354(1).

4. Gack MU, Albrecht RA, Urano T, Inn KS, Huang IC, Carnero E, et al. Influenza A virus NS1 targets the ubiquitin ligase TRIM25 to evade recognition by the host viral RNA sensor RIG-I. Cell host & microbe. 2009;5(5):439-49.

5. Zou W, Zhang DE. The interferon-inducible ubiquitin-protein isopeptide ligase (E3) EFP also functions as an ISG15 E3 ligase. The Journal of biological chemistry. 2005;281(7):3989-94.

6. Nakasato N, Ikeda K, Urano T, Horie-Inoue K, Takeda S, Inoue S. A ubiquitin E3 ligase Efp is up-regulated by interferons and conjugated with ISG15. Biochemical and biophysical research communications. 2006;351(2).

7. Catic A, Fiebiger E, Korbel GA, Blom D, Galardy PJ, Ploegh HL. Screen for ISG15-crossreactive deubiquitinases. PloS one. 2007;2(7).

Reviewer #2: (Please note that all line numbers indicated below refer to the “tracked-changes” version of the manuscript).

1. Can the authors provide the results of statistical comparisons for the data in Figure 2F? The absence of any statistical test is conspicuous given that all other panels in every other figure include statistical comparisons.

2. Line 343: Please do not delete the word “with.” It is critical that the reader understands that VP35 was immunoprecipitated with TRIM25 (in the presence of NP), in reference to the left panel of Figure 5A. Removing the word changes the meaning of the sentence to imply that VP35 immunoprecipitated TRIM25, which is true, but only in the context of the right panel of Figure 5A. Given the discrepancy between the left and right panels, clarity is important here.

3. Line 363: The SPRY and RING/SPRY mutants do not appear to lose their interaction with NP. Instead, the interaction seems to be heavily reduced. The authors should revise this line to make the nuance clear. Similar revisions should be applied to Lines 402 and 583.

4. Lines 404: Similar to the above point, the 7KA mutation has not “lost” its interaction with NP. Instead, like the SPRY mutant, the interaction is reduced. Please revise this sentence (as well as line 583) to make the nuance clear.

5. Did the authors conduct Western blots to evaluate protein expression and pull-down for the RNA IP assays depicted in Figure 7? The amount of RNA detected is likely directly related to the amount of protein precipitated, so it is important to know that sufficient and equal levels of protein were expressed and precipitated.

6. Lines 468 and 477: The authors may want to avoid the use of the term “infection” in this context. While the intent may be clear to some readers, others may not fully grasp that trVLPs are inherently not infectious. Moreover, this particular experiment should not be construed as a replacement for conducting confirmatory experiments with authentic EBOV.

Reviewer #3: (No Response)

PLOS authors have the option to publish the peer review history of their article (what does this mean?). If published, this will include your full peer review and any attached files.

Reviewer #1: No

Reviewer #2: No

Reviewer #3: No
---

## [Editor Report · Decision Letter 2]

18 Apr 2022

Dear Dr Galao,

We are pleased to inform you that your manuscript 'TRIM25 and ZAP target the Ebola virus ribonucleoprotein complex to mediate interferon-induced restriction' has been provisionally accepted for publication in PLOS Pathogens.

Best regards,

Alexander Bukreyev, Ph.D.

Associate Editor

PLOS Pathogens

Christopher Basler

Section Editor

PLOS Pathogens

Kasturi Haldar

Editor-in-Chief

PLOS Pathogens

orcid.org/0000-0001-5065-158X

Michael Malim

Editor-in-Chief

PLOS Pathogens

orcid.org/0000-0002-7699-2064
---

## [Editor Report · Acceptance letter]

4 May 2022

Dear Dr Galão,

We are delighted to inform you that your manuscript, "TRIM25 and ZAP target the Ebola virus ribonucleoprotein complex to mediate interferon-induced restriction," has been formally accepted for publication in PLOS Pathogens.

Best regards,

Kasturi Haldar

Editor-in-Chief

PLOS Pathogens

orcid.org/0000-0001-5065-158X

Michael Malim

Editor-in-Chief

PLOS Pathogens

orcid.org/0000-0002-7699-2064